# A seven-transmembrane methyltransferase catalysing N-terminal histidine methylation of lytic polysaccharide monooxygenases

Tanveer S. Batth [1,5] ✉, Jonas L. Simonsen [1,2,5], Cristina Hernández-Rollán[3,5], Søren Brander[4], Jens Preben Morth [2], Katja S. Johansen[4], Morten H. H. Nørholm [3] ✉, Jakob B. Hoof [2] ✉ & Jesper V. Olsen [1] ✉

Lytic polysaccharide monooxygenases (LPMOs) are oxidative enzymes that help break down lignocellulose, making them highly attractive for improving biomass utilization in industrial biotechnology. The catalytically essential N-terminal histidine (His1) of LPMOs is post-translationally modified by methylation in filamentous fungi to protect them from auto-oxidative inactivation, however, the responsible methyltransferase enzyme is unknown. Using mass-spectrometry-based quantitative proteomics in combination with systematic CRISPR/Cas9 knockout screening in *Aspergillus nidulans*, we identify the N-terminal histidine methyltransferase (NHMT) encoded by the *gene* AN4663. Targeted proteomics confirm that NHMT was solely responsible for His1 methylation of LPMOs. NHMT is predicted to encode a unique seven-transmembrane segment anchoring a soluble methyltransferase domain. Co-localization studies show endoplasmic reticulum residence of NHMT and co-expression in the industrial production yeast *Komagataella phaffii* with LPMOs results in His1 methylation of the LPMOs. This demonstrates the biotechnological potential of recombinant production of proteins and peptides harbouring this specific post-translational modification.

Protein methylation is a post-translational modification (PTM) widely utilized to regulate DNA transcription via the modification of histones at lysine and arginine residues in eukaryotic organisms, constituting the primary component of the so-called "histone code"[1]. Histidine methylation of proteins was recently described in mammalian cells targeting highly abundant actin proteins and its primary function has been suggested to moderately regulate actin filament[2]. More recently, the human methyltransferases METTL9 and METTL18 have demonstrated histidine methylation activity. METTL9 was shown to mediate ubiquitous histidine methylation in mammalian proteomes[3], whereas METTL18 was shown to be a histidine-specific methyltransferase that

specifically targets 60 S ribosomal protein L3 and affects ribosome biogenesis and function[4]. Histidine N-methyltransferase activity has developed on both SET[2] and seven-β-strand domain containing(4) class of methyltransferases.

Histidine methylation[5,6] on fungal lytic polysaccharide monooxygenases (LPMOs) is particularly unique as this methylation occurs only on the N-terminal histidine that is part of the conserved active site[7]. LPMOs are a class of oxidative enzymes found widely in nature. They have broad substrate specificity towards complex polysaccharides, including lignocellulose[8,9]. Cellulose-specific LPMOs are used commercially for the conversion of plant biomass to biofuels[10].

[1]The Novo Nordisk Foundation Center for Protein Research, University of Copenhagen Denmark, Copenhagen, Denmark. [2]Department of Biotechnology and Biomedicine, Technical University of Denmark, Kongens Lyngby, Denmark. [3]The Novo Nordisk Foundation Center for Biosustainability, Technical University of Denmark, Kongens Lyngby, Denmark. [4]Department of Geosciences and Natural Resource Management, University of Copenhagen, Frederiksberg, Denmark. [5]These authors contributed equally: Tanveer S. Batth, Jonas L. Simonsen, Cristina Hernández-Rollán. ✉e-mail: t.batth@cpr.ku.dk; morno@dtu.dk; jblni@dtu.dk; jesper.olsen@cpr.ku.dk

However, N-terminal histidine methylation is only observed in LPMOs expressed by filamentous fungi. A defining feature of all LPMOs is their N-terminal histidine, which is crucial for copper binding and enzyme activity. It is this N-terminal histidine residue that is methylated on its imidazole nitrogen (τ-methylation of Nε2, Fig. 1A). This PTM prevents protonation of the His1 side chain, and it has been demonstrated that it protects the critical active site of the enzyme from oxidative damage[11]. Although X-ray crystallography has shown methylation of the N-terminal histidine in several fungal LPMOs[8,12], the methyltransferase responsible for this modification remains unknown.

In this study, we used state-of-the-art quantitative proteomics in combination with a CRISPR/Cas9-based gene knockout strategy to identify and characterize a previously undescribed N-terminal histidine methyltransferase (NHMT) in the filamentous fungus *Aspergillus nidulans*. We demonstrate that this enzyme is solely responsible for the unique His1 methylation of LPMOs in *A. nidulans*. The methyltransferase is predicted to contain a unique seven helical transmembrane domain and a soluble catalytic domain.

## Results

### Shortlisting N-terminal histidine methylation candidates using bioinformatics and a quantitative proteomics screen

To identify and shortlist NHMT candidates, a large-scale quantitative proteomics screen was performed to identify differentially expressed cellular proteins of cells grown on different carbon sources (Fig. 1B). This strategy was based on the hypothesis that NHMT candidates are co-expressed with methylated LPMO substrates, specifically growth conditions in which the LPMO is needed for polysaccharide conversion, such as when *A. nidulans* is grown with cellulose as an additional carbon source.

Differential protein expression analysis of *A. nidulans* cells grown with the different carbon sources was accomplished through a combination of single-shot label-free quantification (LFQ) and tandem mass tag (TMT)-labelled multiplexed in-depth quantitative mass-spectrometry. Additionally, we performed deep proteome sequencing for *A. nidulans*, identifying almost 7000 protein groups and generating the largest dataset of filamentous fungal proteins to date (Supplementary Data 1).

Over 200 histidine methylation sites were identified in the proteomics dataset, four of which were localized to N-terminal histidines on four different proteins (Supplementary Data 2). MaxQuant software[13] was utilized for the identification of methylated histidine residues within peptides by allowing a variable methyl modification (mass addition of 14.0156 Da) on histidine residues, thereby enabling the Andromeda search algorithm to look for this mass shift within the peptide fragment spectra (MS/MS). Furthermore, a methylated histidine-specific diagnostic immonium ion of m/z 124.0869 was defined to assist in the high-confidence localization of methylated histidines[14]. The four UniProt identifiers found to contain N-terminally histidine methylated proteins were C8V530, Q5B1W7, Q5AU55, and Q5B428. C8V530 (gene ID AN10419), is classified as a member of the Auxiliary Activity Family 9 (AA9) of lytic polysaccharide monooxygenases (LPMOs, Supplementary Table 1)[15–17]. Q5B1W7 (AN5463) plays a crucial role in starch degradation[18], and similarly classified as AA13 family of copper-dependent LPMOs shown to be active on starch[19] (Supplementary Table 2). Q5AU55 (AN4702) is structurally similar to an AA11-type LPMO from *A. oryzae*. Lastly, Q5B428 (AN8175) is highly homologous to an LPMO-like protein from *Laetisaria arvalis* (termed LaX325), part of the recently defined protein family "X325" that are found to be widespread in the fungal kingdom but have deviated evolutionary and do not perform oxidative cleavage of polysaccharides[20]. All four of these LPMO-related proteins contain a secretion signal peptide prior to the N-terminal histidine of the processed proteins.

To identify NHMT candidates, relative protein abundances in cells grown on glucose-containing media (minimal media with glucose or potato dextrose) were compared with those in cells grown with the addition of cellulose. More specifically, we utilized a two-sided t-test with permutation-based false discovery rate (FDR) of 0.05 on log2-fold changes of protein intensity values between growth conditions to determine statistically significant protein abundance differences. This was visualized in a volcano plot of the log2-fold protein changes between two conditions on the x-axis and t-test based statistics on the y-axis; we focused on proteins with putative methyltransferase activity that exhibited differential abundances when grown on different media (Fig. 1C). To bioinformatically determine whether a protein is a methyltransferase in the *A. nidulans* (*Emericella nidulans*) UniProt protein sequence database, we utilized a combination of Interpro[16], Pfam[17], PROSITE[21] and gene ontology protein sequence annotations that predicted S-adenosyl-L-methionine (SAM) dependent methyltransferases, domains, activity, or other related SAM annotations. From this, we manually curated 225 potential methyltransferases in the *A. nidulans* FASTA protein sequence database (Supplementary Data 3). The proteomics analysis of *A. nidulans* identified 120 of the 225 candidate methyltransferases, and 41 of these displayed statistically significant higher abundance when *A. nidulans* cells were cultured on the growth media containing cellulose compared to only glucose or potato dextrose (Fig. 1C, Supplementary Data 4). Seven of the 41 methyltransferases were eliminated due to their prediction (based on combined annotations) as methyltransferases with oxygen as the acceptor atom instead of nitrogen. Additional methyltransferases were eliminated due to the existence of homologs in *Saccharomyces cerevisiae* or *Komagataella phaffii* as these organisms are incapable of performing the N-terminal histidine modification of LPMOs. Moreover, phylogenetic analysis revealed that 19 of the remaining methyltransferases appeared to be unique among fungi known to encode for LPMOs (Supplementary Data 4). To identify the hitherto unknown NHMT enzyme responsible for N-terminal methylation of LPMOs in filamentous fungi, systematic CRISPR/Cas9-based gene knockout screening was carried out on the 19 candidates in *A. nidulans* (Fig. 1D). Another five manually curated methyltransferases that lacked homologs in *Saccharomyces cerevisiae* or *Komagataella phaffii* were included. Of the 24 candidates, 22 were successfully knocked out and assayed for NHMT activity.

### CRISPR/Cas9 knockout screening of 22 methyltransferase candidate genes coupled with targeted MS analysis of LPMO methylation status

To precisely and efficiently knock out each of the 22 genes by CRISPR/Cas9, we designed guide RNAs for each candidate and used a Cas9-expressing *A. nidulans* strain deficient in the error-prone non-homologous end-joining DNA repair, ensuring high-fidelity genome editing through homologous recombination[22,23]. CRISPR/Cas9-mediated gene knockouts were confirmed by diagnostic PCR and to analyse the effect of the individual knockouts on N-terminal histidine methylation, we developed a targeted proteomics assay based on parallel reaction monitoring (PRM) to specifically monitor and quantify the native N-terminally histidine methylated *A. nidulans* peptide sequences identified in the proteome analysis (Fig. 2A). Of the four N-terminal histidine methylated protein sequences detected in the large-scale proteomics screening, we ultimately selected the most abundant His1 methylated peptide corresponding to ([meth]HTVIVYPGYR) from the uncharacterized X325 LPMO-like protein Q5B428 (encoded by the gene AN4702). We selected this protein because its N-terminal histidine methylated peptide ([meth]HTVIVYPGYR) was reproducibly detected in rapid single-shot PRM-MS proteomics analysis (Fig. 2B). We simultaneously targeted the unmethylated peptide (HTVIVYPGYR) in the same PRM assay. The unambiguous observation of the non-methylated Q5B428 N-terminal peptide was required to constitute a NHMT hit in individual knockout strains (Fig. 2B). This quantitative PRM analysis revealed that 21 of the

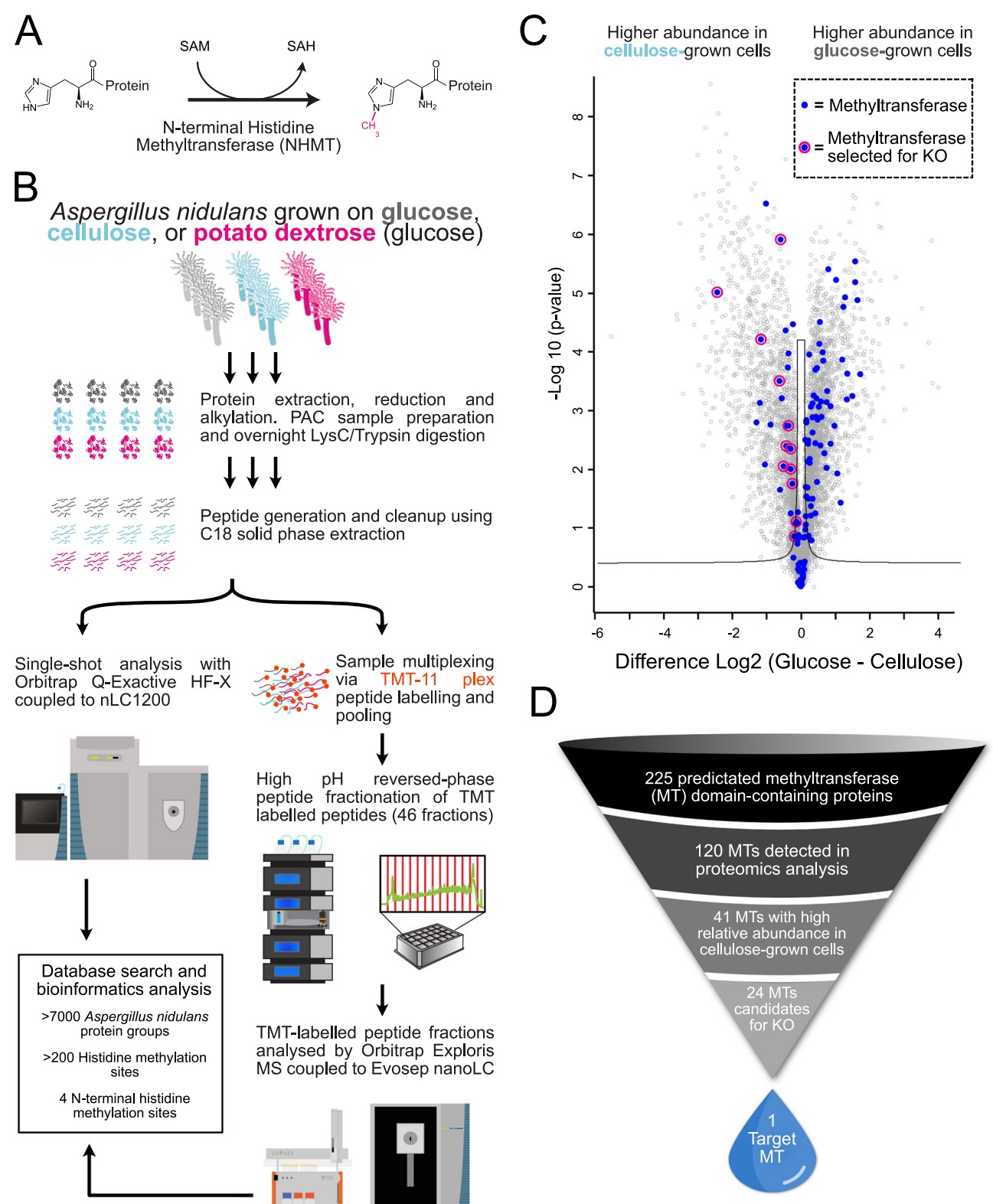

**Fig. 1 | Illustration of the workflow in the NHMT identification process.**
**A** Mechanism of N-terminal histidine methylation via the transfer of a methyl group from S-adenosylmethionine (SAM). **B** *A. nidulans* cells were grown on different *carbon sources* followed by protein extraction and LysC/Trypsin digestion. The peptides generated were analysed using an Orbitrap-based mass spectrometer. **C** Volcano plot showing the protein log2 fold change abundance differences versus their −log10-transformed t-test significance values (from TMT experiment) of cells grown in medium with cellulose versus cells grown in a medium with glucose. All identified methyltransferases (blue dots) and methyltransferases (blue dots with pink outline) selected for knockout from this dataset are illustrated. **D** Graphical illustration of the process leading to the knockout(KO) candidates followed by the identification of the NHMT.

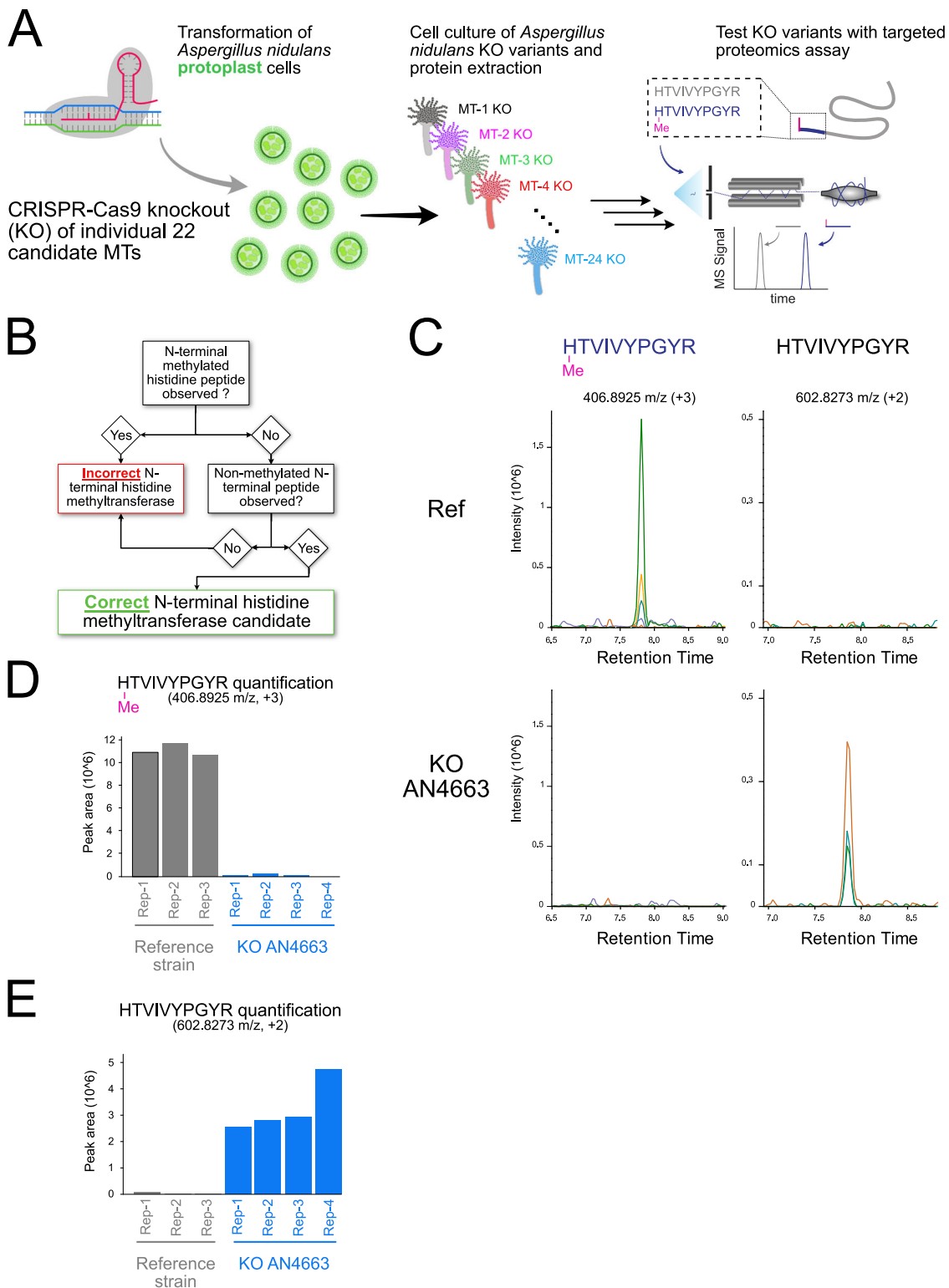

**Fig. 2 | CRISPR/Cas9 knockout and proteomics strategy for identification of the NHMT. A** Schematic representation of the CRISPR/Cas9 knockout library generation and subsequent targeted proteomics assay for identification of the NHMT. **B** Decision flowchart describing the experimental requirements from the PRM assays to nominate a MT to a possible NHMT. **C** PRM chromatogram of N-terminal histidine methylated and unmethylated HTVIVYPGYR peptide in reference strain (top) and AN4663 knockout strain (bottom). **D** The relative quantification of the N-terminal methylation histidine peptide ([meth]HTVIVPGYR) for the reference strain, the knockout candidate AN4663. **E** Relative quantification as in (**D**) of the unmethylated counterpart peptide HTVIVPGYR.

22 methyltransferase knockouts did not affect the methylation state of the targeted X325 protein (Fig. 2A, Supplementary Fig. 1, Supplementary Data 5). As we were unable to test 2 knockout candidates for activity, we cannot say with certainty that they would not have influenced the methylation status of N-terminal histidine residues. Nonetheless, the knockout of the gene encoding for AN4663 (corresponding to the UniProt identifier Q5B467 abolished N-terminal histidine methylation on the monitored methylated peptide and

conversely, the unmethylated form was observed for the first time (Fig. 2B–D).

## AN4663 is the specific N-terminal histidine methyltransferase encoding gene in *A. nidulans*

In silico sequence analysis of the 558-amino-acid-long AN4663 protein using PSIPRED[24] and the transmembrane protein topology prediction tool MEMSAT[25] predicted an N-terminal seven-transmembrane (TM) domain up to amino acid position 215, followed by a soluble methyltransferase-active domain (216-558) at the C-terminal (Fig. 3A and Supplementary Fig. 2). Despite low amino acid identity (22.6%), sequence alignment of AN4663 C-terminal domain (236-493) to the SpdS-like domain (236-493) of the human eukaryotic elongation factor 1 alpha N-terminal methyltransferase (METTL13)[26] identified a conserved glutamic acid at position 340 (E340) which is required for the binding of the co-substrate S-adenosylmethionine (SAM) but not the catalytic activity of methyltransferases[27]. Activity of NHMT was confirmed by site-directed mutagenesis of this glutamic acid to an alanine (E340A) in *A. nidulans*. This E340A mutation resulted in the complete loss of N-terminal histidine methylation, similar to AN4663 gene knockout (Supplementary Fig. 3, Supplementary Data 5). Furthermore, chimeric constructs consisting of an mRFP fluorescence tag fused to the C-terminal of the NHMT greatly diminished N-terminal histidine methylation but not when mRFP was fused to the N-terminus of NHMT (Supplementary Fig. 3, Supplementary Data 5), suggesting that the C-terminal mRFP tag interferes with the NHMTs catalytic activity or alternatively impair correct subcellular localization. Collectively, these data confirm that AN4663, henceforth denoted as *nhmT* when referring to the gene locus, encodes the specific and sole NHMT enzyme in *A. nidulans* responsible for N-terminal histidine methylation of co-expressed LPMOs.

## Structural prediction and sequence analysis supports NHMT activity and predicts a unique transmembrane domain

The AlphaFold2 predicted model of the NHMT structure proposes an N-terminal transmembrane domain (residues 1-215) and a soluble region containing the MTase domain (residues 216-558), complimenting the PSIPRED analysis (Fig. 3B). The predicted model of the soluble MTase domain suggests that NHMT is a member of the 7BS family of methyltransferase with a predicted substrate-binding cavity that is large enough to accommodate the co-substrate SAM with a potential entry pathway that could theoretically allow access for the substrate N-terminal histidine (Fig. 3B, C). Although the AlphaFold2 predicts the structure with high confidence (Supplementary Fig. 4), it is only a prediction and therefore caution must be taken as the limited predictive modeling is done without substrates or co-factors.

AN4663 is annotated as member of the spermidine synthase (SpdS) family of proteins based on sequence homology[28]. This is to be expected as spermidine synthases and methyltransferases have similar protein structures, sequence features, and catalyse the transfer of a methyl group to their respective substrates[29]. As these annotations are linked to enzyme activity based on observed studies, proteins with similar sequences yet distinct activities can be mis-annotated due to a lack of experimental evidence. Sequence alignment of the NHMT catalytic domain using BLAST revealed a single amino acid variance within a conserved motif that could separate the NHMT catalytic domain from spermidine synthases. Specifically, spermidine synthases require either an aspartic acid (D) or glutamic acid (E) in the GxG(D/E)G motif to bind decarboxylated SAM within the extended catalytic pocket (Fig. 3C, D, Supplementary Fig. 5D)[30–32]. However, sequence alignment of the NHMT catalytic pocket revealed that isoleucine at residue position 322 (I322) replaces the D or E in this motif (Fig. 3C, D, Supplementary Fig. 5).

Lastly, interrogation of the NHMT transmembrane domain (1-215) sequence and predicted structure suggests a unique 7TM domain.

Searching the AlphaFold2 predicted 7TM domain model against the DALI server[33] only resulted in low scoring matches to proteins without 7TM domains (Supplementary Fig. 6C). UniProt BLAST search of the 7TM domain (1-215) resulted in 452 matches to homologous 7TM domains containing proteins exclusively to organisms within the filamentous Ascomycota subphylum pezizomycotina expect for one unclassified fungal organism (Fig. 3C, Supplementary Fig. 5B). This analysis revealed ~97% of the matches are single copy genes within their respective organisms. Moreover, all of the 452 matching transmembrane domain sequences were linked to soluble domains that contain a highly conserved segment 389-394 uniquely found in proteins with the NHMT 7TM domain (Fig. 3B, Supplementary Fig. 6B).

## NHMT resides in the endoplasmic reticulum and methylates N-terminal histidines of secreted LPMOs

The LPMOs identified to date are typically secreted outside of the cell by their host organism. Furthermore, all four proteins detected with N-terminal histidine methylation in our proteomics dataset are encoded with a signal peptide, indicating processing through the endoplasmic reticulum (ER). To determine its role in the histidine methylation capacity of NHMT, we examined the importance of its putative NHMT seven-transmembrane region (positions 1-224) by generating a truncated variant NHMT$_{225-559}$ that was lacking the first 224 amino acids. Expression of this variant resulted in elimination of the N-terminal histidine methylation with concomitant appearance of the unmodified version as measured by the PRM assay, indicating that the transmembrane region is critical for its activity (Fig. 4A, B, Supplementary Data 5). We also performed cellular co-localization imaging by fluorescence microscopy using the functional chimera with an mRFP tag fused to the N-terminal of full length NHMT alongside a positive ER marker, the fluorescent organelle probe DiOC$_6$[34] (Fig. 4C). We utilized cytosolic mRFP as a negative ER membrane control (Fig. 4D), and an mRFP tagged mannosyltransferase (AN10118, UniProt entry C8VRA6) involved in protein glycosylation was used as a positive ER localization control[35]. The imaging analysis revealed high co-localization of all mRFP-tagged NHMT constructs with the ER membrane control protein, supporting subcellular localization of NHMT in the ER (Fig. 4C, D).

## Recombinant methylation of LPMOs by NHMT in the yeast *Komagataella phaffii*

To test NHMT activity and determine its biotechnological potential for recombinant N-terminal histidine methylation, we co-expressed NHMT together with a fungal AA9A LPMO from *Lentinus similis* (LsAA9A, UniProt accession A0A0S2GKZ1 or a bacterial AA10 LPMO from *Thermobifida fusca* (TfAA10A, UniProt accession Q47QG3 in the yeast protein production host *Komagataella phaffii* (formerly *Pichia pastoris*), which does not present any endogenous N-terminal histidine activity. The full-length *nhmT* sequence from *A. nidulans* was codon-optimized and expressed without introns in *K. phaffii*. We tested three different secretion peptide sequences for LPMO translocation through the ER using LsAA9A as the methylation substrate (Fig. 5A). These were the native LsAA9A signal peptide, the α-amylase (Amy$^{SP}$) secretion sequence from *A. niger* termed "Amy"[36], and the alpha-mating factor (α-MF) secretion signal leader peptide from *S. cerevisiae*[37,38]. LsAA9A expression and activity were confirmed both with and without co-expression of NHMT (Fig. 5B). LsAA9A was secreted into the supernatant with all three signal peptides (Supplementary Figs. 7–9). However, correct signal peptide processing was only observed and confirmed by LC/MS/MS when using the native or the Amy signal peptide (Supplementary Data 6, Supplementary Fig. 10). We were able to conclusively confirm the N-terminal histidine methylation of LsAA9A upon co-expression with NHMT by using tandem mass spectrometric sequencing. The histidine methylated N-terminal LsAA9A peptide was identified with high confidence and complete amino acid

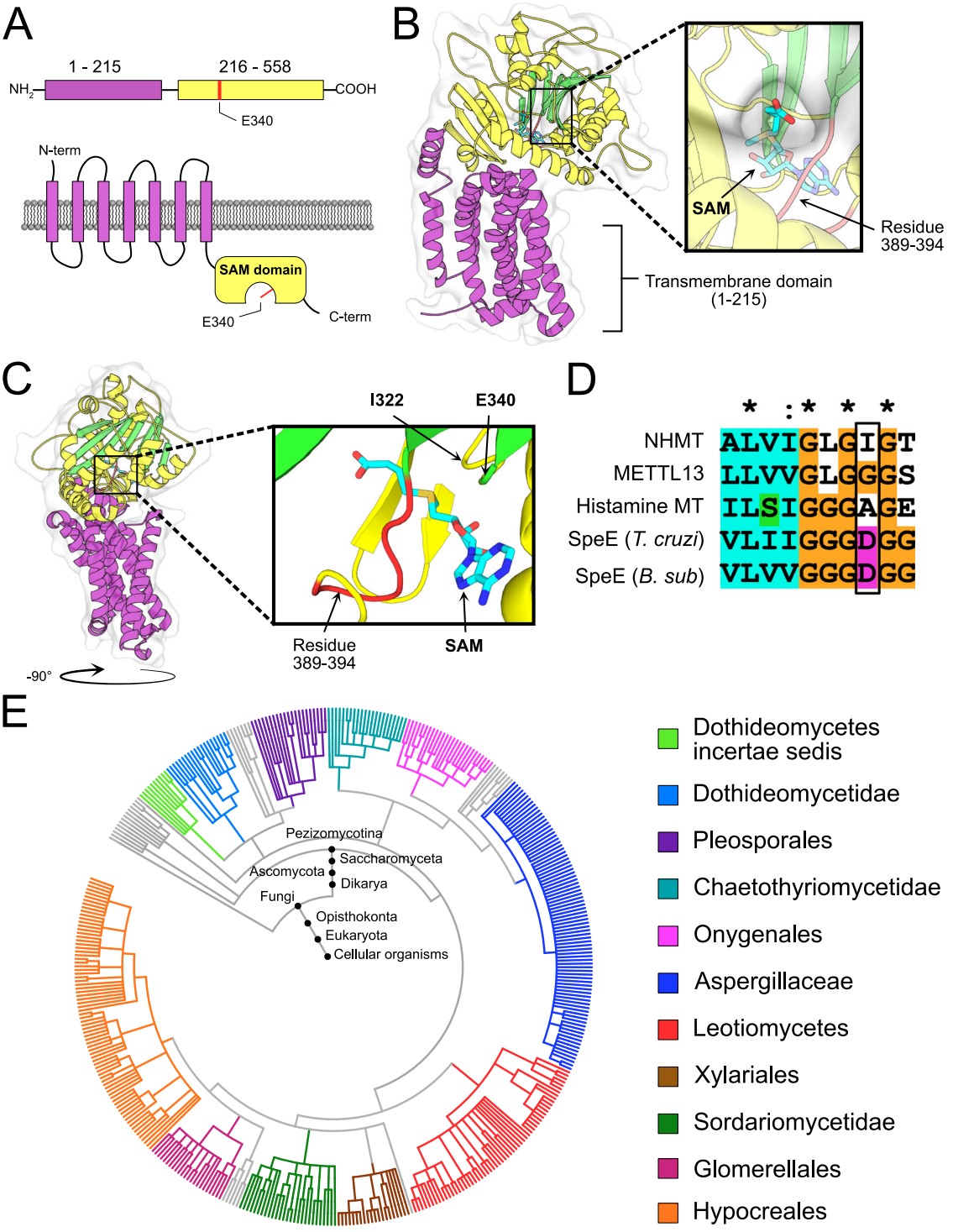

**Fig. 3 | Domain, structure, and phylogenetic prediction from the sequence of NHMT. A** Illustration of the NHMT candidate protein structure predicted by PSIPRED, showing the transmembrane domains (pink) and the soluble catalytic C-terminal (blue), where the targeted mutation E340A is shown in red. **B** The AlphaFold2 model of NHMT predicts the structural domain elements: 7TM (purple) and the soluble methyltransferase containing domain in yellow. The 7-beta strands of the MTase domain are highlighted in green. A close up of SAM moiety in the binding pocket is shown and residues 389-394 highlighted in red. **C** The same structure rotated −90° relative to the viewpoint displayed in B. The SAM binding pocket is zoomed in to highlight the SAM moiety alongside residues I322 and E340. Residues 389−394 are highlighted in red. **D** Alignment of representative sequences, focused on the spermidine synthase motif. (Human eEEF1 n-terminal methyl transferase (METTL13), Human histamine n-methyl transferase, *Trypanosoma cruzi* spermidine synthase (SpeE *T. Cruzi*) and *Bacillus subtilis* spermidine synthase (SpeE *B. sub*). The residues corresponding to I322 are marked with a black box. The lack of an acidic group in this position shows that NHMT is not a spermidine synthase (also see Supplementary Fig. 5C). **E** Phylogenetic analysis of organisms containing a similar NHMT 7TM (1-215) domain. phyloT V2 (https://phylot.biobyte.de/) was used to generate a phylogenetic tree from the taxonomy report of the NHMT 7TM BLAST results. The phylogenetic tree was visualized and major genera highlighted in different colors using the iTOL[60] tool.

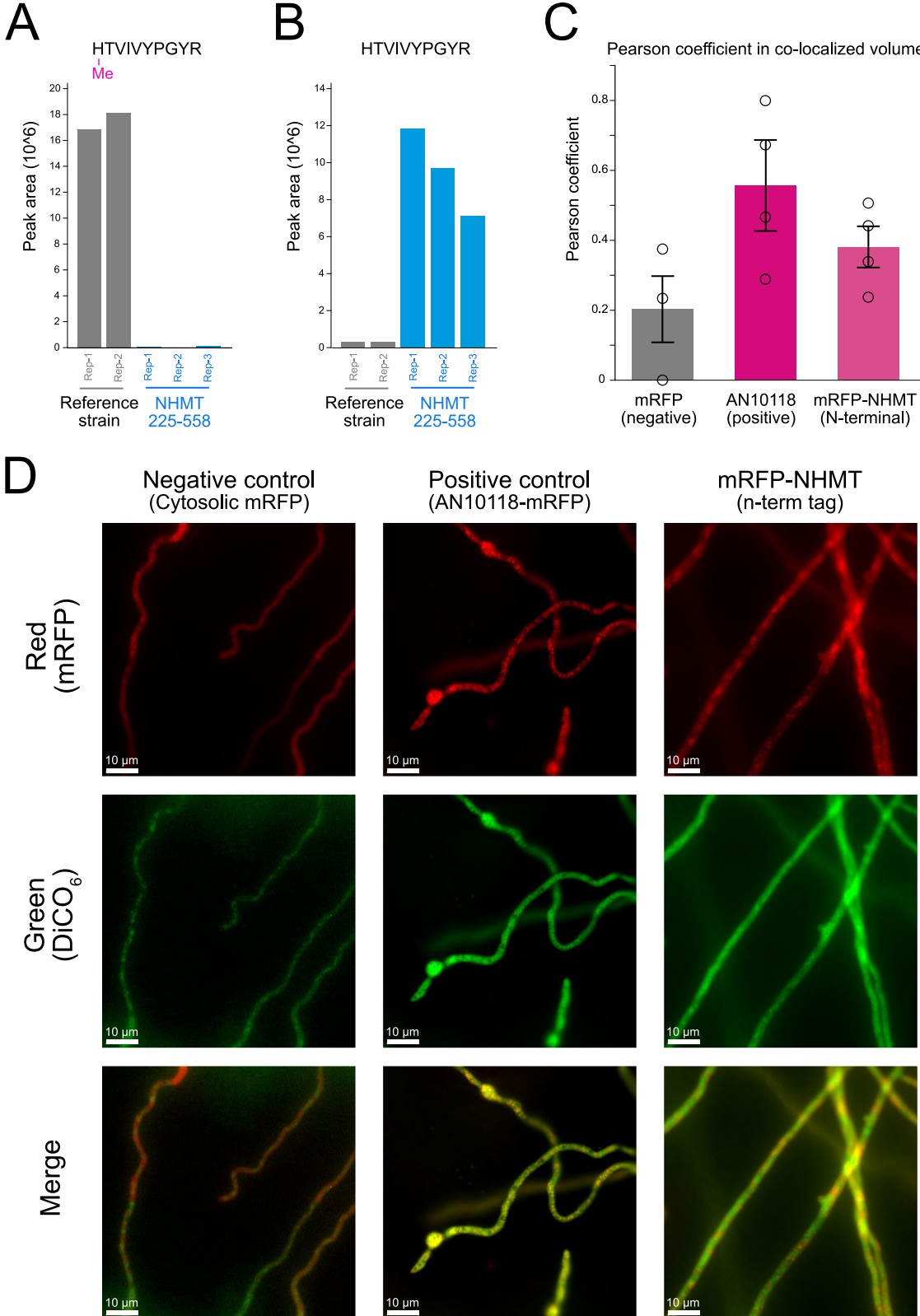

**Fig. 4 | Impact of NHMT truncation on activity and NHMT ER localization.**
**A** PRM quantification of methylated HTVIVPGYR due to truncation of the trans-membrane domain of NHMT. **B** PRM quantification of unmethylated HTVIVPGYR after NHMT truncation. **C** Bar plot showing the mean microscopy quantification of NHMT ER co-localization across different strains with biological replicates ($n \geq 3$) as measured by the Pearson coefficient in co-localized volume of *A. nidulan* filaments. Error bars indicate the standard error of the mean (SEM) of ER localization. **D** Co-localized image of negative control cytosolic mRFP, positive control endogenous mRFP tagged ER protein C8VRA6 and mRFP-NHMT (N-terminal tagged) expressing cells. Two different channels: red (mRFP), green (DiCO$_6$), and merged are shown, yellow indicates co-localization in the merged channel.

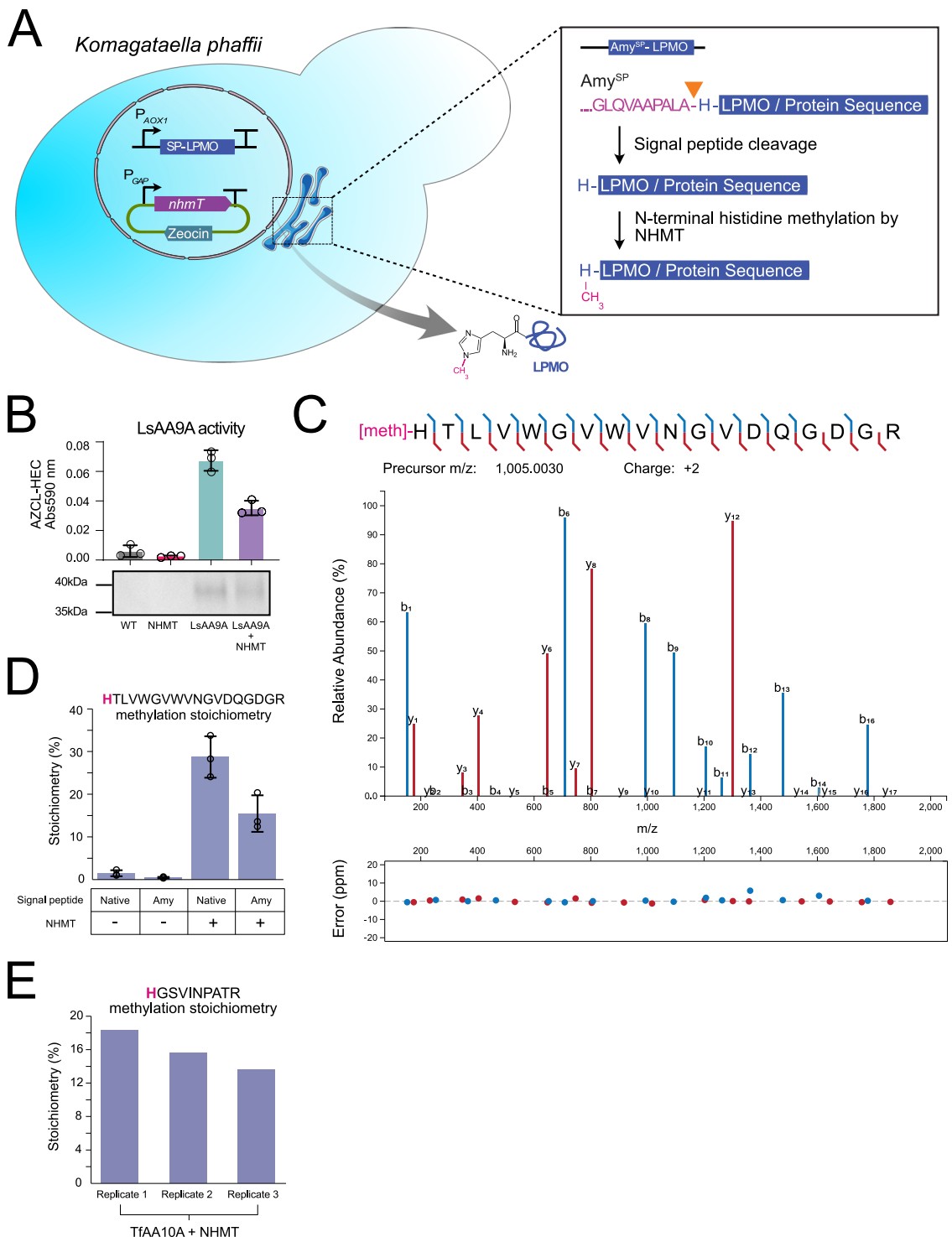

**Fig. 5 | Co-expression strategy of *nhmT* and recombinant N-terminal histidine methylation. A** Schematic of heterologous *nhmT* co-expression in *K. phaffii* for generating recombinant proteins. A DNA construct encoding an LPMO is inserted into the genome and expressed from the strong methanol-inducible promoter p*AOX* together with *nhmT* expressed using a strong constitutive promoter pGAP from an episomal plasmid. The signal peptide (SP) ensures translocation into the ER prior to secretion into the extracellular medium. Figure was partly generated using Servier Medical Art, provided by Servier, licensed under a Creative Commons Attribution 3.0 unported license. **B** Bar plot showing the mean LsAA9A activity detected (upper graph) with an AZCL-HEC assay (see methods). All samples were analysed in biological triplicates (*n* = 3); error bars indicate standard deviation. SDS-PAGE (lower figure) of the LsAA9A secreted into the supernatant of one of the biological replicates. The assay does not provide quantitative information

regarding the amount of active LPMO, rather the main purpose is to detect the prevalence of LPMO activity. **C** MS/MS spectra annotation[62] of the high-confidence identification of N-terminal histidine methylated tryptic peptide of LsAA9A with full sequence coverage of both y- and b-ion series with low fragment mass error. **D** Bar plot showing the mean N-terminal methylation stoichiometry calculation via precursor quantification of the methylated and unmethylated N-terminal tryptic LsAA9A peptide HTLVWGVWVNGVDQGDGR. LsAA9A is secreted into the medium in the presence or absence of the NHMT. All samples were analysed in biological triplicates (*n* = 3); error bars indicate standard deviations. **E** N-terminal methylation stoichiometry calculation via precursor quantification of the methylated and unmethylated N-terminal tryptic TfAA10A peptide HGSVINPATR upon co-expression of TfAA10A and NHMT in *K. phaffii*.

sequence confidently localizing the site-specific methylation to the N-terminal histidine (Fig. 5C). Interestingly, we did not detect histidine methylation by MS when the signal peptide was not cleaved at the correct position N-terminal to the histidine as in the case of using the α-MF as the signal peptide sequence, which resulted in an N-terminally extended LsAA9A harbouring an additional amino acid N-terminal to the histidine with no detectable methylation. This suggests that N-terminal histidine methylation occurs only after precise signal peptide cleavage. We quantified the degree of N-terminal histidine methylation to be ~30% of the identified tryptic LsAA9A N-terminal peptide when co-expressed with NHMT (Fig. 5C, D). TfAA10A was similarly found to be secreted into the supernatant when the sequence was expressed with the Amy signal peptide (Supplementary Fig. 11). Furthermore, we found an average of 15% N-terminal histidine methylation of TfAA10A (Fig. 5E, Supplementary Data 6). These experiments demonstrate that it is possible to engineer a heterologous expression host for specific methylation of N-terminal histidine residues on different protein targets.

## Discussion

In this study, we have identified the NHMT enzyme, which is encoded by the *nhmT* (AN4663) gene and solely responsible for N-terminal histidine methylation of LPMOs. This was accomplished through a combination of in-depth proteomics, bioinformatics, and systematic CRISPR/Cas9 facilitated gene knockouts. The proteomics analysis identified close to 7000 *A. nidulans* protein groups, generating the largest dataset of filamentous fungal proteins to date. Quantitative proteomics analysis of *A. nidulans* cells grown on different carbon sources further enabled us to shortlist potential NHMT candidates that were subsequently selected for CRISPR/Cas9-mediated gene knockouts. We developed a targeted proteomics assay monitoring the methylation state of N-terminal histidine containing peptides from expressed LPMOs to determine the effect of these knockouts and ultimately identified the putative NHMT candidate.

The NHMT enzyme shares a seven-β-strand fold in the soluble domain similar to human histidine methyltransferases such as METTL9 (https://www.uniprot.org/uniprotkb/Q9H1A3), METTL18 (https://www.uniprot.org/uniprotkb/O95568), and the human N-terminal methyltransferase METTL13 (https://www.uniprot.org/uniprotkb/Q8N6R0). Uniquely, NHMT contains an N-terminal seven-transmembrane domain with little structural similarities compared to known membrane proteins in the PDB database. Based on our experimental data, the transmembrane domain is required for the NHMTs specific activity in the context of N-terminal histidine methylation of secreted LPMOs. The sequence of the 7TM region is unique to filamentous fungi and similar sequences are primarily found as single copies in the genomes of filamentous fungi, suggesting distinct phylogeny of the enzyme.

Loss of the N-terminal histidine methylation due to transmembrane truncation could be due to the loss of co-localization with its substrate proteins that require signal peptide cleavage through the ER for secretion. Co-localization imaging suggested that NHMT is indeed located in the ER, providing further support of this theory. Moreover, it is also plausible that the transmembrane region of NHMT is important for trafficking to the ER membrane, correct protein folding and putative oligomeric assembly[39]. The transmembrane region might harbour sites for lipid specific interactions, which could be necessary for its specificity towards N-terminal histidine residues. Furthermore, we observed a significant reduction (but not a complete loss) of methylation activity upon the addition of an mRFP-tag to the C-terminus of the NHMT, suggesting loss of catalytic function or substrate recognition. This could be due to issues related to protein translocation, insertion, or orientation in the membrane. It is also possible that the mRFP tag could prevent access of substrate proteins to into the

catalytic NHMT pocket, inhibiting the transfer of the methyl group from SAM.

The discovery of NHMT prompted co-expression with LPMOs in *K. phaffii*, leading to the demonstration of recombinant N-terminal histidine protein methylation in a heterologous host that is not a filamentous fungus. Future efforts will be directed towards optimizing the co-expression of the NHMT and its targeted substrates to increase methylation stoichiometry, for example, by testing NHMT homologs from other fungi and integrating the corresponding genes into the *K. phaffii* genome together with different substrate-encoding genes. Another limitation could be due to the limited availability of SAM in the ER, as recent plant research has demonstrated that SAM transporters across the Golgi apparatus are required for the efficient methylation of polysaccharides[40]. Optimization of these transporters in the ER might be similarly required in *K. phaffii* to increase the methylation yields on N-terminal histidine residues of proteins such as LPMOs. The findings presented here demonstrate a potential to generate recombinant proteins with methylated N-terminal histidines, which could enable further investigations into the role of this modification in LPMO function and activity. N-terminal histidine methylation could also have potential applications beyond LPMOs, in the production of different classes of pharmaceutical proteins and peptide hormones such as incretins, where methylation of the terminal histidine of GLP-1 for example, may incur protease protection to plasma dipeptidyl peptidase-4[41,42].

This study also highlights the power of combining large-scale quantitative proteomics with a systematic CRISPR editing screen of specific enzyme classes for discovering novel biology in organisms such as the filamentous fungus *A. nidulans*. Filamentous fungi play a crucial role in the degradation of biopolymers and thereby the recycling of organic matter, enabled by the secretion of a battery of enzymes. These enzymes are best known for their roles in biomass degradation that has been exploited in biotechnological and industrial settings[10]. Exploring and characterizing fungal enzymes will thus affect a broad number of bio-based industries. The results presented here pave the way for further proteomics investigations into the diverse phyla of the fungal eukaryotic kingdom, aimed at the discovery of novel PTMs that may lead to the discovery of unique protein attributes.

## Methods

### Strains and cultivation media

The strain NID2531 (*argB2, veA1, nkuA*Δ, Supplementary Information, Supplementary Table 3) was used as the reference strain and subsequent studies and host for all gene deletions and insertions. The strain contained a mutation in the gene encoding ornithine transcarbamylase (*argB2*) of the arginine biosynthesis pathway, which is utilized as selection in transformations via the auxotrophic growth requirement of arginine. Deletion of *nkuA*Δ eliminates to a large extent, the activity of the non-homologous end-joining DNA repair mechanism and forces repair to take place through homologous recombination to promote gene targeting[23]. NID2531 is derived from the wild-type strain FGSC A4, which was used for the initiating *Aspergillus nidulans* proteome experiment. Spore production for inoculations and strain validation were produced in solid minimal medium (MM: 2% agar, 1% glucose, 1 x nitrate salt solution, 0.001% thiamine, 1 x trace metal solution)[43,44], which was supplemented with 4 mM L-arginine when required.

The parent strain used for engineering and protein production was *Komagataella phaffii* GS115 (Thermo Fisher Scientific, Waltham, MA, USA), which has a mutation in the *his4* gene, rendering the strain unable to produce histidine. All plasmids employed for genomic integration contain the HIS4 gene for complementation of the *his4* gene in the parent strain. Transformants are plated on selective medium lacking histidine.

## DNA fragment and plasmid constructions in *A. nidulans* and *K. phaffii*

Fragments constructed by PCR were made as described previously in ref. 23 and CRISPR-Cas9 vectors were assembled by USER fusion[45]. The vectors for *K. phaffii* expression were constructed by LyGo cloning as described in Hernández-Rollán et al.[36], and by USER fusion as described previously in ref. 46. Primers and synthetic gene fragments were purchased from Integrated DNA Technologies (IDT, Coralville, IA, USA), and are listed in Supplementary Table 6. The synthetic gene AN4663 used for expression in *K. phaffii* is listed in Supplementary Notes 3. All PCR fragments for cloning and plasmids were purified using NucleoSpin Gel and PCR Clean-up kit (Macherey-Nagel) and GenElute™ Plasmid Miniprep Kit (Merck), respectively, and the vector assemblies were confirmed by sequencing (Eurofins Genomics, Germany).

All deletions and codon substitutions in *A. nidulans* were enabled by the oligonucleotide-mediated gene-editing procedure described previously in ref. 23, except for the length of the gene-editing oligonucleotides which was 60 nucleotides instead of 90. All CRISPR/Cas9-vectors and expression vectors were assembled via uracil excision-based cloning (USER), as described previously in ref. 23. All CRISPR/Cas9 vectors were based on pFC331 as the recipient vector[22], and were composed by one or more single guide RNAs flanked by tRNAs. All PCRs for DNA-fragment construction in *A. nidulans* engineering were done as described previously in ref. 23. The expression vector for insertion into integration site 5 (IS5) was established in pU0002 as described previously in ref. 47 with a recreated PacI/Nt.BbvCI USER cassette between the targeting sequences of 1.0 kb matching IS5 for cloning of promoter Ptef1, gene fragment, and terminator Ttef1.

For heterologous expression in *Komagataella phaffii*, genes for LsAA9A and TfAA10A were cloned according to the LyGo cloning protocol and integrated into the genome of *K. phaffii*. The native DNA sequence of LsAA9A followed by a TEV cleavage recognition site and a His purification tag was cloned into three different constructs carrying three different signal peptides for secretion in the yeast *K. phaffii*. The following signal peptides were selected: α-MF signal peptide resulting in the vector named pLyGo-*Kp*-1-LsAA9A, Amy$^{SP}$ signal peptide from *A. niger* resulting in the vector pLyGo-*Kp*-2-LsAA9A, and the native *L. similis* signal peptide of LsAA9A resulting in the vector called pPIC9K-Native$^{SP}$-LsAA9A. The LsAA9A native signal peptide was cloned into the vector pLyGo-*Kp*-2-LsAA9A using USER cloning in which both the forward primer FW-Native$^{SP}$_LsAA9A and reverse primer RV-Native$^{SP}$_LsAA9A contained the native signal peptide, resulting in the vector pPIC9K-Native$^{SP}$-LsAA9A. The native DNA sequence without the native signal peptide of TfAA10A from *Thermobifida fusca* (UniProt: Q47QG3) was ordered as synthetic gene fragment from Integrated DNA Technologies (IDT, Coralville, IA, USA) with SapI restriction enzyme sites compatible with LyGo cloning. The gene fragment was cloned by LyGo cloning into the vector pLyGo-*Kp*-2 with Amy$^{SP}$, resulting in the vector pLyGo-*Kp*-2-TfAA10A. The signal peptide Amy$^{SP}$ was chosen to direct the proteins to the ER as it was predicted to be cleaved at position His1 for both proteins according to SignalP[48]. All vectors were cloned and propagated in *E. coli* DH5a, mini-prepped, and their sequences were confirmed by Eurofins Genomics (Ebersberg, Germany).

The synthetic DNA sequence of AN4663 was purchased from Integrated DNA Technologies (Coralville, IA, USA) without introns codon-optimized and cloned into the episomal vector pBGP1 (a kind gift from Charles Lee)[49] using uracil excision-based cloning as described previously in ref. 46 using the primer pair FW-USER-AN4663-episomal and RV-USER-AN4663-GAP-episomal. The expression of the AN4663 in the episomal plasmid pBGP1 is driven by the strong constitutive promoter pGAP. The resulting plasmid was verified by DNA sequencing.

## Competent cells, transformation, and strain validation

Protoplastation for *A. nidulans* was performed as previously in ref. 50. The transformations were made in gently thawed protoplasts[23]. Each transformation protoplasts were mixed with 1.5 µg of CRISPR vector and used in combination with either 2 µg of a linear double-strand DNA for homologous recombination or 20 µL of 100 µM stock solutions of oligonucleotides in a total volume of 150 µL of PCT buffer (50% w/v PEG8000, 50 mM CaCl2, 20 mM Tris, 0.6 M KCl, pH 7.5). The mix was incubated for 10 min at room temperature, followed by the addition of 250 µl of transformation buffer (1.2 M sorbitol, 50 mM CaCl2·2 H2O, 20 mM Tris, 0.6 M KCl, pH 7.2), and plated on transformation media (1 M sucrose, 2% agar, 1x nitrate salt solution, 0.001% thiamine, 1 x trace metal solution plates). All transformation plates were incubated at 37 °C. Resulting transformants were examined by diagnostic tissue-PCR as described previously in ref. 22. Sanger-sequencing was used for validation of strains with AN4663 complementation, point-mutation, and RFP-tagging.

For *K. phaffii*, all vectors were linearized before genomic integration using the primers Fw-PmeI and Rv-PmeI and the linearized fragments were confirmed by electrophoresis prior to electroporation. PCR fragments were further subjected to PCR clean-up using PCR clean-up kit (Macherey-Nagel™). Electrocompetent cells were produced according to the Invitrogen Pichia Expression Kit (Catalogue no. K1710-01). In detail, a starting culture of the desired strain was grown in 5 mL yeast extract peptone dextrose (YPD) medium at 30 °C overnight. Next day, 0.1 mL was used to inoculate a fresh culture of YPD and incubated at 30 °C overnight until an optical density measured at a 600 nm wavelength (OD$_{600}$) of 1.3 was obtained. On the following day, the culture was centrifuged at 1500 g at 4 °C and the cells resuspended in 500 mL ice-cold water. The process was repeated with 250 mL ice-cold water, and once more with 20 mL of an ice-cold sorbitol solution (1 M). After final centrifugation, the cells were resuspended in 1 M sorbitol solution to a final volume of 1.5 mL. Approximately 1 µg of PCR product was used in combination with 80 µL of freshly produced electrocompetent cells (GS115) for electroporation. The cells and the DNA were incubated on ice for 5 min prior to electroporation. For electroporation, a 0.2 cm gap sterile electroporation cuvette was used, and the cells were electroporated using the following parameters: (voltage (V) 1500, capacitance (µF) 25, and resistance (Ω) 200). Right after electroporation, 1 mL of 1 M ice-cold sorbitol was added to the cuvette, and the entire cell pulsed was then plated on minimal dextrose medium (MD), and incubated upside down at 30° for 3–4 days until colonies had formed. The identity of the transformants was confirmed by colony PCR. Colonies grown on MD plates were picked and resuspended in 50 µL of 20 mM NaOH and boiled at 99 °C for 25 min. The mixture was centrifuged for 1 min at 11,000 g to remove cellular debris, and 5 µL of the supernatant was used as a template for the PCR reaction using OneTaq® 2X Master Mix with Standard Buffer (New England Biolabs) primers Fw-AOX1 and Rv-AOX1.

Correct clones producing LsAA9A and TfAA10A were selected for small-scale (in 24-well plates) screening of protein production, and the best producers were then used to make a second batch of electrocompetent cells for the introduction of the episomal plasmid containing AN4663. Selected strains of LsAA9A and TfAA10A expression were made electrocompetent using the method described above, and 1 µg of the episomal plasmid pBGP1-AN4663 was electroporated as described above. After electroporation, 1 M ice-cold sorbitol solution was added to the cuvette, and the mixture was recovered at 30 °C for 2 h with vigorous shaking. The entire culture was then plated on selective YPD plates containing 100 µg/mL Zeocin™ and incubated for up to a week at 30 °C until colonies appeared.

## Co-expression of LsAA9A and TfAA10A together with AN4663

Single colonies of the desired strain were used to inoculate 10 mL of Buffered Glycerol Complex Medium (BMGY) (Pichia Expression Kit,

Life Technologies, Carlsbad, CA, USA) and grown at 28 °C for two days with shaking at 250 rpm. From the saturated culture, a starting culture with an OD600 of 1.0 was prepared in 25 mL buffered methanol complex (BMMY) medium, sometimes supplemented with Zeocin (100 μg/mL) for the strains carrying the pBGP1 plasmid. Baffled glass flasks were used, and the cultures were incubated at 28 °C with shaking at 250 rpm. Expression was maintained by the addition of 1% methanol every day for an additional four days. $OD_{600}$ measurements were carried out daily and the cultures were plated on YPD-agar with 100 μg/ml Zeocin at the end of expression to check for the presence of the pBGP1 plasmid containing AN4663. On the final day, the cells were collected by centrifugation at 5000 g for 20 min, and the supernatant was subjected to filter sterilization. To check for the secretion of LsAA9A and TfAA10A, 5 μL of the supernatant was loaded into a 4−20% Mini-PROTEAN-TGX gel (BioRad, Hercules, CA, USA), run at 165 V for 50 min. After the run, gels were stained with InstantBlue Protein Stain (Expedeon Inc., Inc., San Diego, CA, USA) for one hour and destained overnight with demineralized water.

## AZCL-HEC assay

The activity of secreted LsAA9A from *K. phaffii* was determined using a azurine cross-linked hydroxyethylcellulose (AZCL-HEC) chromagenic assay that enables rapid detection of enzymatic activity on polysaccharides[51]. 1 mg/mL AZCL-HEC substrate (Megazyme, County Wicklow, Bray, Ireland) was mixed with with 1 mM ascorbic acid (Sigma-Aldrich, Saint Louis, MO, USA), 100 μM copper sulphate (Sigma-Aldrich), and the volume was adjusted with 100 mM sodium acetate (pH 5) (Sigma-Aldrich). 100 μL of the secreted LsAA9A sample was mixed with 400 μL of the AZCL-HEC substrate reaction and incubated at 50 °C with shaking at 1500 rpm for 1 h. The samples were centrifuged to remove the AZCL-HEC substrate, and the absorbance was measured at 590 nm.

## Fluorescence microscopy and image analysis

Fresh *A. nidulans* spores suspensions (10 μL of ~$10^5$ spores per mL) were inoculated on glass slides with 0.5 mL solid MM (as described above) with 4 mM L-arginine and incubated for 20 h in petri dishes in micro-perforated bags at 37 °C. A cover slide and immersion oil were applied and the ER was labelled with 10 μM 3,3′-Dihexyloxacarbocyanine iodide ($DiOC_6$) for 20 min. Images of *A. nidulans* were acquired with a Leica DMI6000 widefield microscope, equipped with an 63 × 1.40 OIL HC PL APO objective and operated with LAS X (version 3.3.3) software. Three images of mRFP tagged proteins, green mitochondrial membrane dye ($DiOC_6$) and bright field were acquired for each sample. Collected images were processed using the Imaris software version 9.8.2 (Bitplane AG, Zürich / Oxford Instruments, Abingdon, Oxfordshire, England). Background subtraction was performed followed by manual segmentation of *A. nidulans* filaments and saved as surfaces for each image and co-localization of mRFP and $DiOC_6$ labelled components were determined using Imaris Coloc tool. Intensity threshold of 1000 and 600 were used respectively for the mRFP (red) and $DiOC_6$ (green) channels in all images.

## Proteomics sample lysis and preparation

All fermentations for proteomics analysis were inoculated with a fresh spore suspension in 20 mL liquid media to a concentration of $10^6$ spores per mL in falcon mini bioreactors. Three different liquid media for *A. nidulans* were utilized for the fermentations, these include potato dextrose media containing 39 g/L potato dextrose, minimal media with 1% glucose, or cellulose minimal media (0.5% cellulose, 1% glucose). All media contained 1x nitrate salt solution, 0.001% thiamine, and 1x trace metal solution supplemented with 4 mM L-arginine. Fermentations were carried out for 72 h at 37 °C at 45° degree angle with 180 RPM shaking. Biomass was filtered and biomass were washed with Phosphate-buffered saline and frozen at −80 °C. Five mL of *A. nidulans*

cell biomass was pelleted by centrifugation and washed with cold PBS. Lysis buffer (2% Triton X-100, 1% SDS, 100 mM NaCl and 10 mM EDTA) was added to each sample and vortexed, followed by incubation at 95 °C for 10 min and then vortexed again. 1.5 mL of the resulting solution was transferred to 2 mL bead beating tubes containing 1.4 millimetre Zirconium oxide beads. The tubes were subjected to beating using a Precellys 24 homogenizer (Bertin Technologies, Montigny-le-Bretonneux, France) at 6800 rounds per min for 30 s, followed by sample cooling for 60 s; this was repeated 4 times for a total of 5 times. The resulting liquid was transferred to new 1.5 mL tubes and sonicated using a 2 millimetre sonication tip at 100% power, with 3 s ON and 1 s OFF, for a total sonication time of two min. The resulting liquid was centrifuged at 20,000 g for 10 min at 4 °C, and the resulting supernatant transferred to new 1.5 mL tubes. The protein concentration in the samples was determined using a tryptophan assay[52].

Cysteine residues of the resulting protein lysate were reduced with 5 mM tris(2-carboxyethyl)phosphine and alkylated with 5 mM chloroacetamide for 30 min at room temperature. Protein lysate was prepared for protease digestion with protein aggregation capture (PAC)[53] using magnetic hydroxyl beads. Once the proteins had aggregated on the beads, 50 mM HEPES buffer (pH 8.5) was added. This was followed by the addition of Lys-C protease at a ratio of 1:200 (to total lysate protein) and digestion at 37 °C for 4 h, followed by the addition of trypsin at a ratio of 1:50, and digestion at 37 °C overnight. Samples were acidified to quench the protease reaction by the addition of trifluoroacetic acid (TFA) to a final concentration of 2%.

The resulting peptide mixture was prepared for downstream LC/MS/MS analysis with hydrophobic solid-phase extraction using C18 Waters Sep-Paks (Milford Massachusetts, USA). Briefly, Sep-Paks were prepared by the sequential addition of acetonitrile and 0.1% TFA solution, followed by the addition of the acidified peptide mixture. The samples were then washed with 0.1% TFA and eluted into new tubes using 50% acetonitrile solution with 0.1% TFA. Acetonitrile was evaporated from the peptide solution using an Eppendorf Concentrator Plus speedvac operated at 60 °C. Dried peptides were reconstituted in 0.1% TFA solution, and the peptide concentration was determined using a Thermo Fisher NanoDrop spectrophotometer.

For samples analysed by in-gel proteomics analysis, sample preparation was performed as previously described in ref. 54.

## TMT labelling

Peptides were prepared for labelling using the PAC protocol described above. Briefly, 20 μg of protein was protease-digested on-bead overnight in 50 mM HEPES buffer (pH 8.5). The resulting peptide mixtures were transferred to new tubes and acetonitrile was added to a final concentration of 50%. Tandem mass tag 11-plex reagents (Thermo Fisher Scientific, Waltham, Massachusetts, USA) were added to individual sample and incubated for 60 min at room temperature. The resulting reaction was quenched by the addition of 5% hydroxylamine for 15 min, followed by the addition of TFA to a final concentration of 1%. The labelled samples were pooled and mixed, followed by the evaporation of acetonitrile with a speedvac (see above). The resulting peptide pellet was reconstituted in 50 mM ammonium bicarbonate buffer (pH 8.5) and prepared for offline fractionation.

## High pH reversed-phase offline peptide fractionation

A mixture of 200 μg of the peptides reconstituted in 50 mM ammonium bicarbonate buffer (pH 8.5) was injected, fractionated and collected using a Thermo Scientific UltiMate 3000 high-performance liquid chromatography (HPLC) system. The system was operated at 30 μL/min, and peptides were separated on a Waters Acquity CSH C18 1.7 μm 1 × 150 mm C18 column. Aqueous buffer A (5 mM ammonium bicarbonate) and buffer B (100% acetonitrile) were used for the gradient, which consisted of increasing B from 8 to 28% in 62 min, followed by an increase to 60% B, and another ramp to 70%, where it was

maintained for 8 min, followed by ramp back down to 8% B, where it was maintained for 10 min. For TMT-labelled peptides, the gradient was initially increased to 30% B, followed by a ramp to 65% B and rapid increase to 80% B, where it was maintained for 7 min. The total fractionation time for both methods was 87 min, and a total of 46 fractions were collected.

Formic acid (10%) was added to each fraction, and the samples were dried to completeness using the speedvac. Dried fractions were reconstituted in 40 μL 0.1% formic acid and 20 μL of each fraction was loaded onto Evotips (EvoSep, Odense, Denmark) according to the manufacturer's instructions, and prepared for MS analysis.

### Liquid chromatography coupled to mass spectrometry

Single-shot label-free quantification (LFQ) analysis of protein digests from *A. nidulans* cells grown on different carbon sources were analysed using an EASY-nLC 1200 system (Thermo Fisher Scientific) coupled to the MS with 0.1% formic acid as buffer A and 80% acetonitrile with 0.1% FA as buffer B. 500 ng of peptide mixture was injected onto a homepacked C18 column (15 cm × 75 μm inner diameter) packed with 1.9 μm C18 beads (Dr. Maisch GmbH, Entringen, Germany). The peptide mixture was separated using a gradient of 0 to 25% B in 60 min, after which it was increased to 40% B in 15 min. The gradient was further increased to 80% B in 5 min, where it was held for an additional 5 min, followed by a ramp down to 5% B in 3 min where it was held for 2 min for equilibration. The total runtime was 90 min. A column temperature of 50 °C was applied during all the runs using a column oven (PRSO-V1, Sonation, Biberach, Germany).

An EvoSep One nano-LC system (EvoSep, Odense, Denmark) was coupled to MS for peptide sequencing analysis of individual fractionated samples (TMT labelled and unlabelled) and PRM analysis. Two methods were utilized for all samples: either the 60 samples per day, resulting in a total runtime of 21.5 min, or 30 samples per day, with a total runtime of 44 min. Buffer A, consisting of 5% acetonitrile with 0.1% formic acid, and buffer B, consisting of 0.1% formic and 99.9% acetonitrile, were used for separation. A 15 cm × 150 μm inner diameter homepacked column with 1.9 μm C18 beads was heated to 60 °C, as described above.

### Mass spectrometric data acquisition

The samples were analysed on either a Orbitrap Q-Exactive HF-X or Orbitrap Exploris 480 mass spectrometer (Thermo Fisher Scientific) operated in positive mode with 2 kV spray voltage and a capillary temperature of 275 °C. An MS1 resolution of 120,000 and an MS2 resolution of 30,000 (with an automated gain control (AGC) fill time of 54 s) were used for data-dependent acquisition (DDA). A normalized collisional energy (NCE) of 28 and a quadrupole isolation window of 1.3 m/z were used for LFQ samples, while corresponding values of 35 NCE and 0.8 m/z isolation window were used for TMT-labelled samples.

Targeted analysis was performed using parallel reaction monitoring (PRM) with the Orbitrap Exploris 480 mass spectrometer. An MS1 scan with a resolution of 120,000 was performed prior to an unscheduled product ion scan that cycles through an inclusion list of peptide mass-to-charge (m/z) ratios provided for targeted analysis. Typically, two charge states for each peptide were targeted for PRM analysis in order to increase the specificity. An isolation width of 1.3 m/z, an MS2 resolution of 60,000, a NCE of 30, and maximum injection time of 100 ms were used for each target.

### Mass spectrometry data analysis

All DDA data were searched using MaxQuant software[13]. *A. nidulans* samples were searched against the UniProt *Emericella nidulans* (taxonomy ID 227321, strain FGSC A4 / ATCC 38163 / CBS 112.46 / NRRL 194 / M139) reference proteome. Signal peptides predicted by SignalP[48] were removed from the leading sequence of all signal peptide containing proteins in the FASTA file prior to searching. Files were searched with "Trypsin/P" as the protease specificity, with a maximum of two missed cleavages, fixed carbamidomethyl modification on the cysteines, and variable modifications of oxidation (methionine), acetylation (protein N-terminal), methylation (histidine), phosphorylation (serine, threonine, and tyrosine), and deamidation (asparagine, and glutamic acid) were utilized. Methylated histidine-specific diagnostic ion of 124.0869 m/z was added to the modification within MaxQuant software for high-confidence localization of methylated histidines[14]. The search mass tolerance was first set to 20 ppm, followed by 4.5 ppm after main search recalibration. The mass tolerance for fragment ions was set to 20 ppm. An Andromeda score cut-off of 40 was utilized with a 1% false discovery rate at all levels (peptide spectral matches, peptides, and proteins).

PRM and stoichiometry analysis were performed with the Skyline software[55]. A spectral library was constructed in Skyline from MaxQuant-processed results from label-free samples to assist peak identification. The PRM results were quantified by summing the fragment peak areas (Supplementary Data 5). Summed MS1 peak areas of the three isotopic parent peptide masses were utilized for quantitative comparison in the stoichiometry analysis.

### Downstream bioinformatics and statistical analysis

MaxQuant-processed results were analysed using the Perseus software[56]. LFQ intensities reported in the proteingroups.txt output from MaxQuant were used for the quantitative analysis of label-free samples analysed with LC/MS/MS. Contaminant and reverse hits were removed from the list and the values were log2 transformed. Protein groups were removed if not observed in at least two replicates in one condition. Missing values were imputed using a width of 0.3 and downshift of 1.8. To determine regulated proteins, a two-sided t-test was performed using a permutation-based false discovery rate cut-off of 0.05 as truncation for significant hits. For TMT-labelled samples, reporter intensities were utilized instead of LFQ. Contaminants and reverse hits were similarly removed, and quantile normalization was performed on the reporter intensities followed by log2 transformation. A two-sided t-test was performed as described above to determine significant differences between proteins for LFQ and TMT experiments.

### Alphafold2 modelling

NHMT *Aspergillus nidulans* is a putative enzyme predicted to originate from the spermine/spermidine synthase family protein (AFU_orthologue; AFUA_5G08500). The sequence was subjected to modelling using the AlphaFold2 server at (https://colab.research.google.com/github/phenix-project/)[57] and predicted a model of both the soluble methyltransferase and transmembrane domains based on the confidence score given in an average local Distance Difference Test (lDDT)[58] and plotted in Supplementary Fig. 4. Further structural analysis and SAM docking into the NHMT active site was performed in PyMol for Molecular visualization and RMSD calculation (The PyMOL Molecular Graphics System, Version 2.4.0 Schrödinger, LLC). SAM molecule was taken from Protein Data Bank structure 3BWC[59] and fit into the AlphaFol2 predicted NHMT structure with a single sculpting cycle within PyMol.

### Structural comparison

The soluble methyltransferase containing domain of NHMT (residues 215 to 558) was utilized as search model against the DALI server[33] with a Z score above 10 to identify structural homologous with close relatives of the spermidine syntheses in particular a spermidine synthase from *Trypanosoma cruzi* (PDB ID 4YUX)[31]. The transmembrane domain predicted structure (1-215) was similarly searched against the DALI server.

## Phylogenetic analysis and visualization

The sequence of the NHMT 7TM domain (residues 1-215) was used as input for UniProt BLAST (https://www.uniprot.org/blast) and searched against the uniprotkb_refprotswissprot database using the BLOSUM-45 matrix and an exp cutoff at 1e-4. The taxonomy identifiers from the BLAST results were used as input for phyloT V2 (https://phylot.biobyte.de/) to generate a phylogenetic tree. The phylogenetic tree was visualized and edited using Interactive Tree of Life v6 (iTOL, https://itol.embl.de/)[60].

## Reporting summary

Further information on research design is available in the Nature Portfolio Reporting Summary linked to this article.

## Data availability

The mass spectrometry proteomics data have been deposited to the ProteomeXchange Consortium via the PRIDE[61] partner repository with the dataset identifier PXD037734. All strains, vectors, fragments, primers, and DNA sequences used in this study are listed in supplementary information (Supplementary Tables 3–6, and Supplementary Notes 1–3). Source data are provided with this paper.

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

## Acknowledgements

We would like to acknowledge Antonios Georgantzoglou and Jutta Maria Bulkescher for their assistance in fluorescence microscopy and image analysis. We thank Charles C. Lee for providing us with the episomal plasmid pBGP1 for episomal expression in *K. phaffii*. We also acknowledge Maria Alvarez Arevalo for assistance in constructing *A.nidulans* mutants. We thank Helle Juel Martens assistance with fluorescence microscopy of *A. nidulans*. This work was supported by the Novo Nordisk Foundation (NNF14CC0001 and NNF17SA0027704).

## Author contributions

K.S.J. proposed the study. T.S.B., J.B.H., and J.V.O. conceived the scientific strategy. T.S.B. designed the proteomic strategy and analysed all proteomics-related samples and corresponding data. J.B.H. initiated the development of A. nidulans CRISPR/Cas9 experiments. T.S.B. and J.B.H. designed the knockout candidate list. J.B.H. and J.L.S. designed the A. nidulans engineering strains and expressed the strains for the proteomics and imaging analysis. C.H.R. and M.H.H.N. contributed to the strain engineering of the knockout strains in A. nidulans, designed and engineered K. phaffii for the methylation of LPMOs. S.B. performed the protein sequence alignments and structural analysis. J.P.M. performed AlphaFold2 predictions and protein modelling. T.S.B. and J.V.O. conceptualized and established the manuscript. All authors contributed to the final version of the manuscript.

## Competing interests

The authors have filed a provisional patent on the utilization of N-terminal histidine methyltransferase.
