## [Peer Review File · Nature Communications]

REVIEWER COMMENTS

Reviewer #1 (Remarks to the Author):

This study reports the identification of the methyltransferase that is responsible for N-terminal histidine methylation (NHMT) of LPMO enzymes in *Aspergillus nidulans*.

The manuscript characterizes this MT enzyme as "elusive" (line 344) and suggests that it is of potential practical utility to determine the source of this NHMT activity because the type of LPMO enzyme that are modified by this methylation are "used commercially for the conversion of plant biomass to biofuels" and the methylation has been proposed to improve the LPMO stability. However, there are LPMOs that lack the methylation and yet still have activity, so the it field does not yet appear to be clear on how useful this methylation actually is.

It is not clear whether the already-in-use commercial LPMOs mentioned in the introduction are themselves methylated at the N-terminal histidine. The practical importance of this work is not yet obvious, but it's a good step to learning more. The hypothesis that this methylation will be useful for biomass conversation is clearly of active interest, and finding the enzyme responsible is a key step towards figuring this out.

The key claim in the paper are that the *A. nidulans* gene *nhmtT* (gene AN4663, corresponding to UniProt Q5B467: (1) methylates N-terminal histidines of native peptides in *A. nidulans*, (2) is required for this methylation since disruption of this gene creates the corresponding unmethylated N-terminal peptides, (3) localizes in the ER where LPMOs are believed to be processes, and that (4) this *A. nidulans* gene could be introduced and co-expressed in *K. Phaffii* with a fungal AA9A LPMO from *L. similis* or a bacterial AA10 LPMO from *T. fusca* to achieve 30% and 15% methylation of those LPMOs respectively.

These claims are well supported by the results. This study does represent a key step towards understanding and controlling NHMT activity in yeast systems and then determining the utility of the methylation of N-terminal histidines of bio-produced LPMOs. The degree of methylation achievable (only reaches 30% so far in this study), and the value of the methylation that does occur, will need to be determined in future work that is of great interest in the field. Again, this work represents a key step towards that goal.

There are a several issues that should be addressed in the reporting of this work:

- There are several places where analyses are mentioned in shorthand, but not really described even in brief. For example, starting on line 108, it is stated that "bioinformatic analysis was performed using....[some sources cited] annotations". The words bioinformatic analysis don't really say anything; it would be helpful rather to provide a few words describing the process by which the "225 predicted putative methyltransferase genes were annotated from theFASTA protain sequence database". Is it the case that the *A. nidulans* proteins were aligned by seq. to the sequences in the listed sources" and then the pre-existing annotations in those sources were in turn related to each of the *A. nidulans* proteins where such a sequence similarity meeting some threshold could be found? The details could be put in methods, but the process that you followed should be described at least minimally in the text for key steps in your study.

- On line 113, it is stated that "We ultimately indentified 120 of the 225 predicted methyltranferases in the *A. nidulans* proteome analysis...." How does this relate the process that identified the 225 putative genes. Were there *A. nidulans* proteins in the "*A. nidulans* FAST protein sequence databases" that were not found in your "proteome analysis" that was based on what? This description is not very useful without some clarification of what data were used and in the main text, at least roughly, how they were used to derive these lists and annotations. Then the methods should give a thorough explanation.

- On line 106, it is stated that "volcano plot analysis" was used. Volcano plots are a data display convention, not an analysis. I assume that the analysis implied was simply computing the log-fold changes between relative protein abundances from the two growth conditions and calculating some T-test values (using what model?). Here there is jargon but not a very clear explanation of exactly what was done. Again, the main text should have the brief description and the methods the details.

- In line 212, it is stated that "Automated annotation of the gene product...". Similar to other points above, since you are, for good reason, questioning the accuracy of the automated annotation, it would be helpful to add a few words describing the basis of this annotation other than describing it simply as "automated". If I understand correctly, the gene annotations that you are discussing are based primarily on homology, and then sets of homologous proteins are linked to an activity based on annotations that come from more detailed prior studies of the actual observed activities of representatives of these sets? So these groupings based on homology and prior knowledge of the activities of individual proteins might sometimes err in the functional annotation by grouping proteins that have seq. resemblance but distinct activity? To put the concern in another way, the paper sometimes assigns proteins to families based on databases that use homology and other times makes the distinction between homology-inferred activity versus empirically observed activity for a given protein. This gets confusing unless you spell it out.

- In lines 224-230, there is an interesting observation about a distinction at a particular residue position for spermidine synthases versus methyl transferases. It could be useful to put some numbers on this with respect to the residues present at this position for proteins with different annotations. At the beginning of the section, it is highlighted that NHMT was MISS-annotated as a spermidine synthase, but here at the end of the section, you use prior annotation to support the importance of this residue for accurate distinctions between spermidine synthases and methyl transferases based also on prior annotation, but presumably annotation that you feel is not all incorrect as was the NHMT annotation? It is complicated to use these database annotations themselves to help illuminate errors in the basis for those very type of annotations. Your deductions look reasonable as best I can discern, but the logic for why you question some of the annotation (within the motif you cite) but then trust and use other annotation (within that same motif!) from the same type of sources deserves a clearer explanation.

- In the CRISPR screen, the KOs are not all validated. Therefore the positive results are more interpretable whereas thenegative results in the assay are more ambiguous cannot be assumed to be confirmation of lack of enzyme activity but could simply be due to a lack of KO such that the target gene was not actually tested. The extensive follow-up in this study clarifies the role of the key hit gene - and suggests that the other candidate genes were inactive - so this lack of KO validation does not really matter for this study in hindsight. But this is not immediately clear from the CRISPR screen itself, so that should be clarified in the reporting of the CRISPR results.

- The introduction is a bit of a grab bag of more and less relevant information. For example, it is noted that LPMOs have been implicated in several fungal, viral and bacterial mediated infections that can be lethal. I see that this statement is a close quote from a sentence in one of the review papers on LPMOs that was cited, but there is not attempt to explain how LPMOs are involved or how this connects the current study.

Reviewer #2 (Remarks to the Author):

This paper by Batth/Simonsen et al describes the discovery of the methyltransferase responsible for methylation of the N-terminal histidine in LPMOs and LPMO-like proteins in fungi. This is an important discovery and the work done to get to this discovery is state-of-the art and elegant. The paper is easy to follow, and the story is nice. I only have minor comments that may improve the paper.

While I enjoyed reading this paper, I became somewhat irritated by the quite many basic grammatical errors (which seem not equally distributed throughout the manuscript and SI). The authors should carefully read through the complete paper and SI and fix the language where needed. Some specific comments follow below.

I find the sequence and structural analysis part of the paper a bit long and a bit speculative in part. Many words are used to describe relatively simple aspects of the study. Grammatical errors are relatively frequent in these parts. Predictions are discussed as if they were hard experimental evidence. AlphaFold is great, but may make (minor) mistakes, especially for proteins composed of both integral membrane and soluble domains. I encourage the authors to critically assess this part of the manuscript to make sure that it is not very long and contains sufficient cautionary notes. See also specific comments below.

Specific comments:

1. Line 88 and elsewhere: I find the use of the term "protein groups" confusing. Is there any serious reason for not just using "proteins"?
2. Line 90: "Over 200 histidine methylation sites were identified in the dataset,". I would like to see some more explanation here for readers not too familiar with proteomics. Please spend a few words on explaining how these were detected. Is the mass shift associated with His methylation sufficiently unique?
3. Lines 91-102: Please polish this section a bit, providing the same type of information for all four proteins. Which AA families do they belong to? The phrasing "of the auxiliary activity family type" is strange. And were these predictions really "based on automatic fold recognition and structure prediction"? This is not really what InterPro and Pfam do. Why are there only supplementary Tables for two of the proteins?
4. Figure 1C: All of the 24 targeted transferases should be labeled and it should be possible to see which one is AN4663.
5. Discussion: Would it possible and meaningful to shortly discuss the possible functions of the other methyltransferases that are upregulated during growth on cellulose?
6. L156-L157: Please rephrase and correct reference to Figure (2B, not 2A).
7. Figure 2D & E: Typo (peptide charges are incorrect)
8. L173-273: As mentioned above, I think there is room for improvement and some shortening in the text on sequence and structural analysis. Furthermore, here and there cautionary notes are needed. For example, Lines 176-180 seem not needed. Why discuss this type of sequence analysis when Alphafold can produce a plausible 3-dimensional model?
9. L182 "with low amino acid homology": replace homology by identity and give the percentage.
10. L194: When mentioning subcellular organization here, please add something like "see below".
11. Figures 3B-D should be moved to the supplementary material (this is just assay data and the nature of the assay is sufficiently illustrated by Fig. 2); Figure 3A can be merged with Figure 4 (or deleted – see above).
12. L222 and elsewhere: glutamine or glutamate?
13. Figure 4B: Please provide information about how SAM was modelled in the Alphafold structure. I would suggest to add a few key residue numbers (e.g. domain borders) in the actual figure. Mark 389-394 (line 269).
14. Legend to Figure 4D: First sentence needs rephrasing. Provide clear and correct information about what was done.
15. L250-253, "The structural model predicts NHMT to contain an intermediate sized substrate-binding cavity required to contain the co-factor SAM and the substrate n-terminal histidine, which is in agreement with its NHMT activity (Figure 4C)." This is not precise, not well phrased and potentially speculative. What is an "intermediate size" and what did the authors do to rationally link this size to the substrate? Do the authors have an idea or proposal regarding how NHMT interact with the substrate?
16. L279-283: Please rephrase text for clarification. The truncation shows that the TM domain is important, but can one really say that this mutant alone shows localization in the ER?

17. Figure 5: Move A and B to the SI. About C & D: exactly which variants are shown where and for what reason? There are five in C and only three in D.
18. L308: For clarity, replace "protein" by "LPMO".
19. Figure 6: L332, the legend should inform about the promoter used for expression of nhmT from the episomal plasmid; "The signal peptide ensures" -> "The signal peptide (SP) ensures";
20. Figure 6B: The activity assay used will, under the conditions used here, not give accurate quantitative information regarding the amount of active LPMO, whereas the Figure may give the impression that the presence of NHMT reduces LPMO activity. To make sure that readers do not get a false impression, please add a cautionary note to the legend explaining that this assay is not much more than a yes/no assay (which is perfectly fine for the purpose of this study)
21. Figure 6C is trivial and confusing at the same time (it seems). This seems pretty standard to me, but I got confused when reading the text. What particular point are the authors trying to make here? Can this Figure be deleted?
22. Lines 370-371: What about issues related to translocation and membrane insertion? "substrate blockage" needs rephrasing.
23. Line 372 and onwards: The methylation "yields" in *Pichia* were not very high and, for now, I slightly doubt the biotechnological use if this system. In this paragraph the authors inform about their intent to improve this. It would be very interesting if there would be some more discussion on what type of improvements could be made. What is the problem?
24. Line 468: What do the numbers 5990 and 5991 refer to?
25. Line 548: explain the abbreviation AZCL-HEC.
26. The legends of Figures S5-S9 needs serious editing. I use the legend of Fig. S5 as an example. It says: "SDS- PAGE of secreted samples into the supernatant from the co-expression of LsAA9A and the methyltransferase AN4663 when the LsAA9A is expressed using the α -MF signal peptide in biological triplicates." That is terrible, "lazy" writing. The information provided in the subsequent sentence between parenthesis varies between the strains and is, thus, inconsistent and confusing (e.g., "LsAA9A secreted into the supernatant"). I do not understand why the arrow indicates an "expected" position in the gel. What is this expectation based on?
27. Table S2. Is this an AA10 or an AA13? Some explanatory text may be useful.
28. Tables S1 & S2. Why are similar Tables not provided for the other two proteins (including the one that was used in the knock out studies aimed at finding the NHMT)?
29. Note S1 and S2: Please indicate if these are the gene sequences as found in the host organism or those synthesized for protein expression in *Pichia*.

Some additional grammatical issues:

- Lines 55-56: Rephrase
- Line 71: fungi-> fungus
- Line 166: "the process required" makes no sense
- Line 247: are -> is
- Line 255-257 & 260-262: Rephrase
- Lines 353 - 360: Bad writing. Polish.

Reviewer #3 (Remarks to the Author):

Response to Author,

Congratulation on your study. I appreciate the effort in detail work, with CRISPR/Cas9 gene knockout, to gain insights into NHMT biological role and location. The research methods used in the study were appropriate and executed well with details description in the supplement section. The manuscript is well written; clear, and easy to understand.

Some minor suggestions:

1. Line 222: add amino acid abbreviations - glutamine 340
2. Line 658: add percentile of acetonitrile in buffer B
3. Author run two method 21.5 vs 44 minutes for peptide sequencing analysis after fractionation. Any recommendations for the reader which method is better?

4. Supplement figure 6 and 7. Author claims that molecular weight is around 25.2kDA, however the bands on the gel are around 37kDA for LsAA9A with or without NHMT.
5. Supplement figure 7 - replicate 1, column 5 (LsAA9A + NHMT), it looks like molecular marker. Please provide better gel.
6. Gene knockout PRM assay supplement figure 1 - combine methylated (609.8351 m/z) and unmethylated (602.8273 m/z) HTVIVYPGYR peptide into one graph.
7. Supplementary Data Set 5, contains PRM quantification used to generate graph in Figure 4A, B. However, figure 4 shows: "Motif and structure and activity prediction from the sequence of NHMT". Please correct.
8. Supplement figure 8 contains two out of three calculations, concentrated-LSAA9_MTn is missing, please provide rational why it was not included.

Reviewer #4 (Remarks to the Author):

This is an interesting work of applying extensive experimental works to identify and characterize a new N-terminal histidine methyltransferase (NHMT) encoded by the gene AN4663 that methylate Lytic polysaccharide monooxygenases (LPMOs). The experiments such as proteomics experiments and gene knockout experiments are well designed and convincingly determine the function and localization of NHMT. The bioinformatics analysis of the protein structure and function such as AlphaFold structure prediction is reasonable. Overall, the methods are solid, and the results are interesting.

Comments for revision:

The bioinformatics methods for predicting the structure of NHMT, searching the predicted structure against known structures, constructing multiple sequence alignments for NHMT, and building phylogenetic trees for NHMT proteins across multiple species are not described in detail. A section describing the bioinformatics in detail should be added into the Materials and Methods Section. The data and scripts of the bioinformatics analysis should be made publicly available to reproduce the analysis.

Point-by-point rebuttal letter to REVIEWER COMMENTS

Manuscript: NCOMMS-22-50384-T (Batth et al.)

Our responses to the **reviewers' comments** are provided below in **blue font**. Key modifications and updates are indicated in the combined manuscript file using track changes.

Reviewer #1 (Remarks to the Author):

This study reports the identification of the methyltransferase that is responsible for N-terminal histidine methylation (NHMT) of LPMO enzymes in *Aspergillus nidulans*.

The manuscript characterizes this MT enzyme as "elusive" (line 344) and suggests that it is of potential practical utility to determine the source of this NHMT activity because the type of LPMO enzyme that are modified by this methylation are "used commercially for the conversion of plant biomass to biofuels" and the methylation has been proposed to improve the LPMO stability. However, there are LPMOs that lack the methylation and yet still have activity, so the it field does not yet appear to be clear on how useful this methylation actually is.

It is not clear whether the already-in-use commercial LPMOs mentioned in the introduction are themselves methylated at the N-terminal histidine. The practical importance of this work is not yet obvious, but it's a good step to learning more. The hypothesis that this methylation will be useful for biomass conversation is clearly of active interest, and finding the enzyme responsible is a key step towards figuring this out.

The key claim in the paper are that the *A. nidulans* gene *nhmtT* (gene AN4663, corresponding to UniProt Q5B467: (1) methylates N-terminal histidines of native peptides in *A. nidulans*, (2) is required for this methylation since disruption of this gene creates the corresponding unmethylated N-terminal peptides, (3) localizes in the ER where LPMOs are believed to be processes, and that (4) this *A. nidulans* gene could be introduced and co-expressed in *K. Phaffii* with a fungal AA9A LPMO from *L. similis* or a bacterial AA10 LPMO from *T. fusca* to achieve 30% and 15% methylation of those LPMOs respectively.

These claims are well supported by the results. This study does represent a key step towards understanding and controlling NHMT activity in yeast systems and then determining the utility of the methylation of N-terminal histidines of bio-produced LPMOs. The degree of methylation achievable (only reaches 30% so far in this study), and the value of the methylation that does occur, will need to be determined in future work that is of great interest in the field. Again, this work represents a key step towards that goal.

We thank the reviewer for the positive comments and the constructive recommendations to strengthen the manuscript. The reviewer is correct to point out that the impact on the utility and "usefulness" of LPMO N-terminal histidine methylation is yet to be determined, and we hope that this work will be crucial in answering this question.

There are a several issues that should be addressed in the reporting of this work:

- There are several places where analyses are mentioned in shorthand, but not really described even in brief. For example, starting on line 108, it is stated that "bioinformatic analysis was performed using....[some sources cited] annotations". The words bioinformatic analysis don't really say anything; it would be helpful rather to provide a few words describing the process by which the "225 predicted putative methyltransferase genes were annotated from theFASTA protein sequence database". Is it the case that the *A. nidulans* proteins were aligned by seq. to the sequences in the listed sources" and then the pre-existing annotations in those sources were in turn related to each of the *A. nidulans* proteins where such a sequence similarity meeting some threshold could be found? The details could be put in methods, but the process that you followed should be described at least minimally in the text for key steps in your study.

We agree with the reviewer that this section lacks clarity and we have revised the statement accordingly. Based on bioinformatics we predicted 225 putative methyltransferases in the FASTA proteome. In the mass spectrometry-based proteomics analysis, we identified 120 of those 225 proteins. We agree that this is a bit confusing and we have changed the sentence to the following for clarity:

"To bioinformatically determine whether a protein is a methyltransferase in the *A. nidulans* (*Emericella nidulans*) UniProt protein sequence database, we utilized a combination of Interpro¹⁵, Pfam¹⁶, PROSITE²¹ and gene ontology protein sequence annotations that predicted S-adenosyl-L-methionine (SAM) dependent methyltransferases, domains, activity, or other related SAM annotations. From this, we manually curated 225 potential methyltransferases in the *A. nidulans* FASTA protein sequence database (Supplementary Table S3)."

- On line 113, it is stated that "We ultimately identified 120 of the 225 predicted methyltransferases in the *A. nidulans* proteome analysis...." How does this relate the process that identified the 225 putative genes. Were there *A. nidulans* proteins in the "*A. nidulans* FAST protein sequence databases" that were not found in your "proteome analysis" that was based on what? This description is not very useful without some clarification of what data were used and in the main text, at least roughly, how they were used to derive these lists and annotations. Then the methods should give a thorough explanation.

We agree with the reviewer that this section lacks clarity and we have adjusted the statement accordingly. We predicted 225 methyltransferases in the FASTA sequence database, and in the mass spectrometry proteomics analysis, we identified 120 of those 225 proteins we predict could be methyltransferases when we searched the collected MS data against the same FASTA sequence database for *A. nidulans*. We agree that this is a bit confusing and we have changed the sentence to the following for clarity:

"The proteomics analysis of *A. nidulans* identified 120 of the 225 candidate methyltransferases, and 41 of these displayed statistically significant higher abundance when *A. nidulans* cells were cultured on the growth media containing cellulose compared to only glucose or potato dextrose (Figure 1C, Supplementary Table S4)."

- On line 106, it is stated that "volcano plot analysis" was used. Volcano plots are a data display convention, not an analysis. I assume that the analysis implied was simply computing the log-fold changes between relative protein abundances from the two growth conditions and calculating some T-test values (using what model?). Here there is jargon but not a very clear explanation of exactly what was done. Again, the main text should have the brief description and the methods the details.

The reviewer is correct to point out that a volcano plot is only a visualization of the statistical tests utilized to determine significantly different protein abundances between the growth conditions. Following the advice of the reviewer, we have changed the text for clarity and added an explanation of the statistical test we utilized.

"More specifically, we utilized a two-sided t-test with permutation-based false discovery rate (FDR) of 0.05 on log₂-fold changes of protein intensity values between growth conditions to determine statistically significant protein abundance differences. This was visualized in a volcano plot of the log₂-fold protein changes between two conditions on the x-axis and t-test based statistics on the y-axis; we focused on putative methyltransferases that exhibited different abundances when grown on different media (Figure 1C)."

- In line 212, it is stated that "Automated annotation of the gene product...". Similar to other points above, since you are, for good reason, questioning the accuracy of the automated annotation, it would be helpful to add a few words describing the basis of this annotation other than describing it simply as "automated". If I understand correctly, the gene annotations that you are discussing are based primarily on homology, and then sets of homologous proteins are linked to an activity based on annotations that come from more detailed prior studies of the actual observed activities of representatives of these sets? So these groupings based on homology and prior knowledge of the activities of individual proteins might sometimes err in the functional annotation by grouping proteins that have seq. resemblance but distinct activity? To put the concern in another way, the paper sometimes assigns proteins to families based on databases that use homology and other times makes the distinction between homology-inferred activity versus empirically observed activity for a given protein. This gets confusing unless you spell it out.

We thank the reviewer for correctly pointing out that gene product annotations are typically based on homology and some prior knowledge, which is used to infer protein function. We agree that there are some nuances in the sequences and consequently this process may not always be accurate. We agree that the sentence is a bit confusing and ambiguous so we have changed the start of the paragraph in order to address these valid concerns to the following:

"AN4663 is annotated as member of the spermidine synthase (SpdS) family of proteins based on sequence homology²⁸. This is to be expected as spermidine synthases and methyltransferases have similar protein structures, sequence features, and catalyse the transfer of a methyl group to their respective substrates²⁹. As these annotations are

linked to enzyme activity based on observed studies, proteins with similar sequences yet distinct activities can be mis-annotated due to a lack of experimental evidence.”

- In lines 224-230, there is an interesting observation about a distinction at a particular residue position for spermidine synthases versus methyl transferases. It could be useful to put some numbers on this with respect to the residues present at this position for proteins with different annotations. At the beginning of the section, it is highlighted that NHMT was MISS-annotated as a spermidine synthase, but here at the end of the section, you use prior annotation to support the importance of this residue for accurate distinctions between spermidine synthases and methyl transferases based also on prior annotation, but presumably annotation that you feel is not all incorrect as was the NHMT annotation? It is complicated to use these database annotations themselves to help illuminate errors in the basis for those very type of annotations. Your deductions look reasonable as best I can discern, but the logic for why you question some of the annotation (within the motif you cite) but then trust and use other annotation (within that same motif!) from the same type of sources deserves a clearer explanation.

We thank the reviewer for highlighting this point. We agree that this dichotomy can cause confusion to the readers: the functional annotations are wrong, but prior annotations also help in determining key residues for specific enzyme activity. Based on the reviewer’s suggestion, we now clarified that there can be sequence homology between similar set of enzymes with slightly different donor and substrate activities. Finding out what sequence features determine these differences for proteins that are very similar can be difficult. Nonetheless, in this case the residue we highlight is within the extended binding and catalytic pocket. However, there are other motifs in the sequence outside of the binding pocket that can also display sequence similarity through which the automated annotations can wrongly annotate gene products. Nonetheless, we were able to pinpoint a probable amino acid residue, which can differentiate NHMT from spermidine synthases. We have also reworded the entire section with cautionary notes, and in this instance for clarity. It now reads as the following:

“Sequence alignment of the NHMT catalytic domain using BLAST revealed a single amino acid variance within a conserved motif that could separate the NHMT catalytic domain from spermidine synthases. Specifically, spermidine synthases require either an aspartic acid (D) or glutamic acid (E) in the GxG(D/E)G motif to bind decarboxylated SAM within the extended catalytic pocket (Figure 3C-D, Supplementary Figure S5D)^{30–32}. However, sequence alignment of the NHMT catalytic pocket revealed that isoleucine at residue position 322 (I322) replaces the D or E in this motif (Figure 3C-D, Supplementary Figure S5).”

- In the CRISPR screen, the KOs are not all validated. Therefore the positive results are more interpretable whereas the negative results in the assay are more ambiguous cannot be assumed to be confirmation of lack of enzyme activity but could simply be due to a lack of KO such that the target gene was not actually tested. The extensive follow-up in this study clarifies the role of the key hit gene - and suggests that the other candidate genes were inactive - so this lack of KO validation does not really matter for this study in hindsight. But

this is not immediately clear from the CRISPR screen itself, so that should be clarified in the reporting of the CRISPR results.

The reviewer is indeed correct to note that not all of the KOs were validated and thus the absence of evidence is not evidence of absence. We have amended the manuscript where we describe the CRISPR results to include a statement clarifying that we are unable to test 2 of the knockouts. Thus we cannot say for certain whether or not these two proteins could have an effect:

“As we were unable to test 2 knockout candidates for activity, we cannot say with certainty that they would not have influenced the methylation status of N-terminal histidine residues. Nonetheless, the knockout of the gene encoding for AN4663...”

- The introduction is a bit of a grab bag of more and less relevant information. For example, it is noted that LPMOs have been implicated in several fungal, viral and bacterial mediated infections that can be lethal. I see that this statement is a close quote from a sentence in one of the review papers on LPMOs that was cited, but there is no attempt to explain how LPMOs are involved or how this connects the current study.

We agree with the reviewer that the statement regarding lethality of LPMOs might not have high relevance to the present study. We have removed the quote regarding infections from the manuscript.

Reviewer #2 (Remarks to the Author):

This paper by Batth/Simonsen et al describes the discovery of the methyltransferase responsible for methylation of the N-terminal histidine in LPMOs and LPMO-like proteins in fungi. This is an important discovery and the work done to get to this discovery is state-of-the-art and elegant. The paper is easy to follow, and the story is nice. I only have minor comments that may improve the paper.

While I enjoyed reading this paper, I became somewhat irritated by the quite many basic grammatical errors (which seem not equally distributed throughout the manuscript and SI). The authors should carefully read through the complete paper and SI and fix the language where needed. Some specific comments follow below.

I find the sequence and structural analysis part of the paper a bit long and a bit speculative in part. Many words are used to describe relatively simple aspects of the study. Grammatical errors are relatively frequent in these parts. Predictions are discussed as if they were hard experimental evidence. AlphaFold is great, but may make (minor) mistakes, especially for proteins composed of both integral membrane and soluble domains. I encourage the authors to critically assess this part of the manuscript to make sure that it is not very long and contains sufficient cautionary notes. See also specific comments below.

We thank the reviewer for the thorough assessment of our manuscript and the constructive feedback. All the authors have carefully gone through another round of editing of the manuscript text to identify and address grammatical errors. We thank the reviewer for highlighting the drawbacks of predictions, and we agree with the reviewer that it is correct to

caution on their overreliance. Consequently, we have shortened the structural analysis section as suggested by the reviewer and added cautionary notes highlighting the limits of some of the predictive analysis.

Specific comments:

1. Line 88 and elsewhere: I find the use of the term “protein groups” confusing. Is there any serious reason for not just using “proteins”?

We thank the reviewer for this key question and the reason for using “protein groups” is due to the nature of the MS-based proteomics technology itself. As we performed “bottom up” proteomics, where proteins are first digested with trypsin into short peptides for sequencing, sometimes it is not possible to assign a single identified peptide to a single protein in the database. This often happens if the peptide sequence matches multiple proteins such as isoforms or proteins with similar sequences. In this case most peptide search algorithms report “protein groups”, where there is not sufficient evidence or unique peptide sequences matching to a single protein. Although a majority of “protein groups” are typically single protein hits, there are number of instances where 2 or more proteins will be grouped as one entry from the same set of identified peptides.

2. Line 90: “Over 200 histidine methylation sites were identified in the dataset,”. I would like to see some more explanation here for readers not too familiar with proteomics. Please spend a few words on explaining how these were detected. Is the mass shift associated with is His methylation sufficiently unique?

The reviewer is correct to point out that adding clarity to how the histidine methylation peptides were identified is important because the localization of the methylation site is critical as other amino acids besides histidine can also be methylated, for example arginines and lysines. We used the Andromeda peptide search engine in the MaxQuant software suite to search the tandem mass spectrometry data. Peptide identifications from this software are considered highly reliable as the search algorithm uses only high scoring matches followed by 1% false discovery rate filtering of all peptides. Furthermore, we added a variable modification of methylation on histidine residues as a search parameter within the software. This means that the software will look for mass shifts among amino residues of fragmented peptides (in the MS/MS scans) that correspond to a methylated histidine. Lastly, when a methylated histidine containing peptide is fragmented in the mass spectrometer (in the MS/MS spectra), the methylated histidine residue generates a very specific immonium ion with a specific mass-to-charge (m/z) as reported by Kapell et al (2021). This “diagnostic ion” can be used to assist in the identification of methylated histidine containing peptides and increase the confidence of the localization. We have added the reference to the MaxQuant software and the utilization of the diagnostic ion for methylated histidines in the main text and changed the sentence to the following:

“MaxQuant software¹³ was utilized for the identification of methylated histidine residues within peptides by allowing a variable methyl modification (mass addition of 14.0156 Da) on histidine residues, thereby enabling the Andromeda search algorithm to look for this mass shift within the peptide fragment spectra (MS/MS). Furthermore, a methylated histidine-specific diagnostic immonium ion of m/z 124.0869 was defined to assist in the high-confidence localization of methylated histidines¹⁴”

3. Lines 91-102: Please polish this section a bit, providing the same type of information for all four proteins. Which AA families do they belong to? The phrasing “of the auxiliary activity family type” is strange. And were these predictions really “based on automatic fold recognition and structure prediction”? This is not really what InterPro and Pfam do. Why are there only supplementary Tables for two of the proteins?

We thank the reviewer for this suggestion and agree that the phrasing could be improved. We have removed the term “auxiliary activity family type” from the text. The author is correct that the InterPro and pFAM predictions are not based on automatic fold recognition and structure prediction and removed the statement. We have reworded this section for clarity. There are supplementary information for the two proteins as those were the only ones that pFam could accurately predict to contain a glycohydrolase domain or classified as AA9 type protein (as in the case for C8V530). We have added more information on the proteins and added 2 extra references to support the statements.

“The four Uniprot identifiers found to contain N-terminally histidine methylated proteins were C8V530, Q5B1W7, Q5AU55, and Q5B428. C8V530 (gene ID AN10419), is a protein predicted to contain a glycoside hydrolase domain by InterPro¹⁵ and Pfam¹⁶ (Table S1) and it is classified as a member of the Auxiliary Activity Family 9 (AA9 LPMO) glycohydrolase by CAZy¹⁷. Q5B1W7 (AN5463) plays a crucial role in starch degradation¹⁸, and similarly contains a glycoside hydrolase domain classified as AA13 family of LPMOs shown to be active on starch¹⁹ (Table S2). Q5AU55 (AN4702) is structurally predicted to be most similar to an AA11-type LPMO from *A. oryzae*. Lastly, Q5B428 (AN8175) is highly homologous to an LPMO-like protein from *Laetisaria arvalis* (LaX325), part of the recently defined protein family “X325” that are found to be widespread in the fungal kingdom but have deviated evolutionary and do not perform oxidative cleavage of polysaccharides²⁰.

4. Figure 1C: All of the 24 targeted transferases should be labeled and it should be possible to see which one is AN4663.

The volcano plot shown is the comparative analysis between the cells grown on cellulose containing media vs glucose from the TMT mass spectrometry experiment. The 24 target candidates were summarized from the comparative analysis of cellulose and glucose, cellulose and potato dextrose in both TMT and label free mass spectrometry experimental analysis (total of 4 different comparisons). In this case, we only displayed the comparative statistical analysis (shown as a volcano plot) from one experiment. From this experiment alone, there are 12 MTases (statistically regulated) that were also among the 24 that were selected for knockouts. There was an overlap of MTases between the comparisons and some of which were only found in one of the analysis as not all of the comparisons resulted in 24 total KO candidates. We did not label the candidates here as it would be visually difficult to show all of the 12 labels in Figure 1C alone, and labeling of Uniport identifiers would not necessarily mean much to most readers. Nonetheless, AN4663 is among the 12 candidates here, and the

decision to not label it was more to keep in line with the manuscript text as the real target is not revealed until Figure 2.

5. Discussion: Would it possible and meaningful to shortly discuss the possible functions of the other methyltransferases that are upregulated during growth on cellulose?

We believe it was beyond the scope of this paper and our expertise to speculate on the numerous other methyltransferases, which were found to be differentially regulated. Nonetheless, the supplementary data and Excel tables include information on all of the differentially abundant proteins in the different conditions, and includes a column indicating whether the protein is a methyltransferase (based on our analysis) in the proteomics data for easy analysis and filtering.

6. L156-L157: Please rephrase and correct reference to Figure (2B, not 2A).

We thank the reviewer for this suggestion, we have changed the figure reference and rephrased the sentence to the following:

“We selected this protein because its N-terminal histidine methylated peptide ([meth]HTVIVYPGYR) was reproducibly detected in rapid single-shot PRM-MS proteomics analysis (Figure 2B). We simultaneously targeted the unmethylated peptide (HTVIVYPGYR) in same PRM assay. The observation of the non-methylated Q5B428 N-terminal peptide was required to constitute a hit in individual knockout strains (Figure 2B).”

7. Figure 2D & E: Typo (peptide charges are incorrect)

The peptide charges are correct. Methylated peptide sequence we monitored ([meth]HTVIVYPGYR) with a charge state of +3 results in a mass-to-charge (m/z) ratio of 406.8925. Multiplying this number by 3 (as that is the charge and multiplication gives the mass) gives a m/z 1218.6 with a charge state of 1, which would have a monoisotopic peptide mass (without any charges) of 1217.6 Da. Similarly, for the unmethylated peptide sequence (HTVIVYPGYR) we monitored the +2 charge state peptide ion which results in m/z of 602.8273. Multiplying this number by 2 gives a 1204.6473 m/z with a charge state of 1. Note that the methylated peptide has higher m/z (at charge state 1) of 14, corresponding to the increase mass due to the methyl group.

8. L173-273: As mentioned above, I think there is room for improvement and some shortening in the text on sequence and structural analysis. Furthermore, here and there cautionary notes are needed. For example, Lines 176-180 seem not needed. Why discuss this type of sequence analysis when Alphafold can produce a plausible 3-deminsional model?

We thank the reviewer for this input and agree that these sections should be reworded. Consequently, we have improved this section and shortened the in-silico and bioinformatics analysis section of the manuscript.

9. L182 “with low amino acid homology”: replace homology by identity and give the percentage.

We have replaced “homology” with “identity” and gave a percentage based on the alignment. The sentence now reads the following:

“Despite low amino acid identity (22.6%), sequence alignment of AN4663 C-terminal domain (236-493) to the SpdS-like domain (236-493) of the human N-terminal methyltransferase METTL13²⁶ identified a conserved glutamic acid at position 340 (E340) which is required for the binding of the co-substrate S-adenosylmethionine (SAM) but not the catalytic activity of methyltransferases²⁷.”

10. L194: When mentioning subcellular organization here, please add something like “see below”.

We have added “(see below)” to the aforementioned line.

11. Figures 3B-D should be moved to the supplementary material (this is just assay data and the nature of the assay is sufficiently illustrated by Fig. 2); Figure 3A can be merged with Figure 4 (or deleted – see above).

Thanks for the suggestion, which we have followed. We have removed figure 3 from the manuscript and moved the assay down to the supplementary information. Figure 3A has been merged with Figure 4 (now the new Figure 3).

12. L222 and elsewhere: glutamine or glutamate?

We thank the reviewer for pointing out this mistake, we have replaced glutamine and glutamate with glutamic acid throughout the text.

13. Figure 4B: Please provide information about how SAM was modelled in the AlphaFold structure. I would suggest to add a few key residue numbers (e.g. domain borders) in the actual figure. Mark 389-394 (line 269).

We have added AlphaFold2 modelling section to the Materials and Methods to provide detailed information regarding the predicted structure. Briefly, SAM molecule from a crystal structure of spermidine synthase was used to fit within the AlphaFold2 predicted NHMT structure. We have updated Figure 4 (now Figure 3) to highlight residues and different part in the predicted structure. We have updated the figure to show two different angles of the structure and the SAM binding pocket and highlighted residues 389-394 as well as I322 and E340. Specifically, this is shown in the new Figure 3B and 3C (see below).

14. Legend to Figure 4D: First sentence needs rephrasing. Provide clear and correct information about what was done.

We have updated the legend to Figure 4D (now Figure 3) for clarity and provided more specific information about the phylogenetic tree and how it was generated in the figure legend. We have additionally added a detailed section in the Materials and Methods titled “Phylogenetic analysis and visualization”. The figure legend for the phylogenetic tree has been updated to the following:

D) Phylogenetic analysis of organisms containing a similar NHMT 7TM (1-215) domain. phyloT V2 (<https://phylot.biobyte.de/>) was used to generate a phylogenetic tree from the taxonomy report of the NHMT 7TM (1-215) pBLAST results. The phylogenetic tree was visualized and major genera highlighted in different colors using the iTOL³⁴ tool.

15. L250-253, “The structural model predicts NHMT to contain an intermediate sized substrate-binding cavity required to contain the co-factor SAM and the substrate n-terminal histidine, which is in agreement with its NHMT activity (Figure 4C).” This is not precise, not well phrased and potentially speculative. What is an “intermediate size” and what did the authors do to rationally link this size to the substrate? Do the authors have an idea or proposal regarding how NHMT interact with the substrate?

We agree with the reviewer that the statement is not precise and speculative. As suggested by the reviewer, we have reduced the length and speculation of the sequence and structural analysis section of the paper. We have rephrased this particular sentence and removed the ambiguous “intermediate size” wording when it comes to describing the substrate binding cavity. There are limits of predicted structural models, and lack of a crystal structure limit our capabilities to confidently propose mechanism of substrate, SAM co-factor, and NHMT interactions. Nonetheless, the predicted structure does enable basic confirmation of the

MTase domain, such as the presence of seven-beta strands and binding pocket where the SAM molecule can be fit. Based on reviewer suggestions we have added cautionary note in the text as we moderated the speculative nature of the structural analysis based only on predicted models. The sentence now has been updated (alongside the entire section) with the following:

“The predicted model of the soluble MTase domain corroborates that NHMT is a 7BS family of methyltransferase with a predicted substrate-binding cavity that is large enough to accommodate co-factor SAM with a potential entry pathway that could theoretically allow access for the substrate N-terminal histidine (Figure 3B).”

16. L279-283: Please rephrase text for clarification. The truncation shows that the TM domain is important, but can one really say that this mutant alone shows localization in the ER?

The reviewer is correct to highlight that the mutation alone cannot discern whether the protein is located to the ER and other explanations are still possible. We have amended this section upon further reflection. We have now changed this particular section from:

“We therefore hypothesized that NHMT must be located in the ER to methylate its substrates. To test this, we generated a truncated variant of NHMT lacking the transmembrane region (positions 1-224); the expression of this variant abolished the N-terminal histidine methylation capacity of the enzyme, indicating that the transmembrane region is critical for its specific activity (Figure 5A, B, Supplementary Table S5).”

To the following:

“To determine its role in the histidine methylation capacity of NHMT, we examined the importance of its putative NHMT seven-transmembrane region (positions 1-224) by generating a truncated variant NHMT₂₂₅₋₅₅₉ that was lacking the first 224 amino acids. Expression of this variant resulted in elimination of the N-terminal histidine methylation with concomitant appearance of the unmodified version as measured by the PRM assay, indicating that the transmembrane region is critical for its activity (Figure 4A-B, Supplementary Table S5).”

17. Figure 5: Move A and B to the SI. About C & D: exactly which variants are shown where and for what reason? There are five in C and only three in D.

We thank the reviewer for their feedback regarding Figure 5. We however disagree that Figure 5A and 5B should be moved to supplementary information. We believe it is an important result that establishes the requirement of the transmembrane domain for successful methylation of substrate N-terminal histidine residues and the data are discussed throughout the text in the results as well as the discussion. We agree with the reviewer about the mismatch of the different variants in Figure 5C and Figure 5D. We showed additional variants because we attempted to increase the expression of mRFP tagged NHMT (at C and N terminus) using a high expression constitutive promoter in integration site 5 (e.g. IS5::Ptef-AN4663-mRFP-Ttef, found in the strain supplementary table) to drive the over-expression of these constructs with the hope of better fluorescence. Ultimately this strategy did not give

significantly higher expression compared to native chromosomal expression where mRFP was inserted in the gene at the locus (mRFP-NHMT, n-terminus). We have removed the extra variants from the figure as they are not discussed in the main text and updated the Pearson correlation in the co-localized volume to reflect this. The new Figure 5C (Figure 4C in the updated version as original Figure 3 was removed based on reviewer suggestion) looks like the following:

18. L308: For clarity, replace “protein” by “LPMO”.

We have replaced “protein” with “LPMO” in this instance.

19. Figure 6: L332, the legend should inform about the promoter used for expression of *nhmT* from the episomal plasmid; “The signal peptide ensures” -> “The signal peptide (SP) ensures”;

We thank the reviewer for noticing this missing information. We have added a description to the legend on the GAP promoter used for *nhmT* expression and changed the sentence to the following:

“A DNA construct encoding an LPMO is inserted into the genome and expressed from the strong methanol-inducible promoter pAOX together with *nhmT* expressed using a strong constitutive promoter pGAP from an episomal plasmid.”

We have also made the changes so that the next sentence reads “The signal peptide (SP) ensures...”

20. Figure 6B: The activity assay used will, under the conditions used here, not give accurate quantitative information regarding the amount of active LPMO, whereas the Figure may give the impression that the presence of NHMT reduces LPMO activity. To make sure that readers do not get a false impression, please add a cautionary note to the legend explaining that this assay is not much more than a yes/no assay (which is perfectly fine for the purpose of this study)

We thank the reviewer for noting this important distinction. The reviewer is correct to point out that the purpose of the assay was more qualitative than quantitative, and we agree that

this distinction is not apparent. We have added a cautionary note in legend to clarify this. Specifically, in the legend for Figure 4B we have changed it to the following:

“B) LsAA9A activity was detected (upper graph) with an AZCL-HEC assay (see methods). All samples were analysed in biological triplicates; error bars indicate standard deviations. SDS-PAGE (lower figure) of the LsAA9A secreted into the supernatant of one of the biological replicates. The assay does not provide quantitative information regarding the amount of active LPMO, rather the main purpose is to detect the prevalence of LPMO activity.”

21. Figure 6C is trivial and confusing at the same time (it seems). This seems pretty standard to me, but I got confused when reading the text. What particular point are the authors trying to make here? Can this Figure be deleted?

The reviewer is correct to point out that the Figure 6C is fairly standard. The main point is to show the actual MS/MS spectra evidence of the identified N-terminally histidine methylated peptide of this LPMO. The figure is directed towards those who are inclined to manually look at the tandem mass spectrometric evidence of the modification. The figure shows that the detected peptide mass-to-charge (precursor m/z as referred to in the figure) matches the expected m/z of that peptide sequence modified with the addition of a methyl group. Furthermore, the figure shows complete peptide backbone sequence coverage using peptide fragmentation spectra of γ -ions (ie. Fragmentation series from c-terminus) and b-ions (fragmentation series starting from the N-terminus). The coverage is only possible if the mass differences between the fragments correspond to the addition of the methylation group on the first histidine in the peptide. For primary novel findings in proteomics, this type of presentation of the primary evidence is customary for proteomics experts as they do not have to download the raw files and look at the MS/MS spectra evidence or reprocess the data to reproduce the primary evidence. Though this is always an option as the raw files are deposited in a public repository alongside the processed data from search algorithms. We have amended the legend for 6C to the following in order to stress the finding:

“C) MS/MS spectra annotation⁴⁰ of the high-confidence identification of N-terminal histidine methylated tryptic peptide of LsAA9A with full sequence coverage of both γ - and b-ion series with low fragment mass error.”

We have also amended the reference to this figure in the main text to stress the high confidence of this identification. Specifically, we have changed the text originally in line 313 – 316 from:

“LsAA9A was secreted into the supernatant with all three signal peptides (Supplementary Figure S5, S6, S7), however correct signal peptide processing was only observed and confirmed with LC/MS/MS when using the native and the Amy signal peptide (Figure 6C, Supplementary Table 6, Supplementary Figure S8).”

To the following

“LsAA9A was secreted into the supernatant with all three signal peptides (Supplementary Figure S5, S6, S7), however correct signal peptide processing was only observed and confirmed with LC/MS/MS when using the native and the Amy signal

peptide (Supplementary Table 6, Supplementary Figure S8). We were able to conclusively confirm the N-terminal histidine methylation of LsAA9A upon NHMT co-expression using mass spectrometry sequencing of the N-terminal peptide which was identified with high confidence and complete sequence coverage corresponding to the site-specific methylation at the N-terminal histidine (Figure 6C)."

22. Lines 370-371: What about issues related to translocation and membrane insertion? "substrate blockage" needs rephrasing.

The reviewer is correct to note that there can be additional issues related to translocation and proper membrane insertion. We have amended this sentence and clarified the phrasing around "substrate blockage". We have changed the lines originally 370-371 from:

"Furthermore, we observed a significant reduction (but not loss) of methylation activity upon the addition of an mRFP tag at the C-terminal of the NHMT, suggesting loss of catalytic function or substrate recognition, via either structure inhibition or substrate blockage."

To the following:

"Furthermore, we observed a significant reduction (but not a complete loss) of methylation activity upon the addition of an mRFP-tag to the C-terminus of the NHMT, suggesting loss of catalytic function or substrate recognition. This could be due to issues related to protein translocation, insertion, orientation in the membrane. It is also possible that the mRFP tag could prevent access of substrate proteins to into the catalytic NHMT pocket, inhibiting the transfer of the methyl group from SAM."

23. Line 372 and onwards: The methylation "yields" in *Pichia* were not very high and, for now, I slightly doubt the biotechnological use if this system. In this paragraph the authors inform about their intend to improve this. It would be very interesting if there would be some more discussion on what type of improvements could be made. What is the problem?

We thank the reviewer for raising this criticism, and we agree that further discussion is warranted. Optimizing and balancing the expression of multiple genes in a cell factory can be a challenging task. We have revised this paragraph to elaborate on the challenges and possible solutions to increase the methylation stoichiometry. Specifically, we have changed the aforementioned paragraph with the following and added a reference from Temple et al, (2022, Nature Plants) to support the discussion:

"...Future efforts will be directed towards optimizing the co-expression of the NHMTs and the targeted substrates to increase methylation stoichiometry, for example by testing NHMT homologs from other fungi and integrating the corresponding genes into the *K. phaffii* genome together with different substrate-encoding genes. Another limitation could be due to the limited availability of SAM in the ER as recent plant research has demonstrated that SAM transporters across the Golgi apparatus are required for the efficient methylation of polysaccharides.⁴² Optimization of these transporters in the ER might be similarly required in *K. phaffii* to increase the methylation yields on N-terminal histidine residues of proteins such as LPMOs"

24. Line 468: What do the numbers 5990 and 5991 refer to?

Numbers 5990 and 5991 are the ID numbers of the primers used to insert the native signal peptide of LsAA9A into the vector pLyGo-Kp-2-LsAA9A. We agree that written as number identifications and as primer description is redundant and can be confusing. We have amended the text and the supplementary information for better clarification from:

“The LsAA9A native signal peptide and cloned into the vector pLyGo-Kp-2-LsAA9A using USER cloning in which both the forward (5990) and the reverse primer (5991) contained the native signal peptide, resulting in the vector pPIC9K-NativeSP-LsAA9A.

The synthetic DNA sequence of AN4663 was purchased from Integrated DNA Technologies (Coralville, IA, USA) without introns codon-optimized and cloned into the episomal vector pBGP1 (a kind gift from Charles Lee) 51 using uracil excision-based cloning as described previously 48 using the primer pair 5701 and 5702.

For *K. phaffii*, all vectors were linearized before genomic integration using the primers 4917 and 4918 and the linearized fragments were confirmed by electrophoresis prior to electroporation. PCR fragments were further subjected to PCR clean-up using PCR clean-up kit (Macherey-Nagel™).

The mixture was centrifuged for 1 minute at 11,000 g to remove cellular debris, and 5 µL of the supernatant was used as a template for the PCR reaction using OneTaq® 2X Master Mix with Standard Buffer (New England Biolabs) primers 4925 and 4926.”

Primers for K. phaffii		
4817	aA_PmeI_F	AAACGCTGCTCTTGGAACTA
4818	aA_PmeI_R	AAACTGTCAGTTTTGGGCCAT
5552	Fw_SapI_AN4663	CGGGTTTGCTCTTCGCATACAACGACCGCACTTAATAATATCATCGC TG
5359	Rv_SapI_AN4663	AAACCCGGGGCTCTTCGTTACCAGCCCTCCCAGACGCG
5990	FW-NativeSP_LsAA9A	ACGGCACTUTCCTTCGTCGCGTCAGCCGCGGCTCATACCTCGTCT GGGGC
5991	RV-NativeSP_LsAA9A	AAGTGCCGUGAGCCCGAGGATGGAGTACTTCATCGTTTGGATCCTT CGAATAATTAGTTGTTTTTGATC
5701	FW-USER-AN4663-episomal	ATTGAACAACUATTTGAAACGATGGCGCCTTTTCGTTCAATTTATG
5702	RV-USER-AN4663-GAP-episomal	AGTTGTTCAAUTGATTGAAATAGGGACAAATAAATTAATTTAAAG TC

To the following:

“The LsAA9A native signal peptide and cloned into the vector pLyGo-Kp-2-LsAA9A using USER cloning in which both the forward primer FW-NativeSP_LsAA9A and the reverse primer RV-NativeSP_LsAA9A contained the native signal peptide, resulting in the vector pPIC9K-NativeSP-LsAA9A.

The synthetic DNA sequence of AN4663 was purchased from Integrated DNA Technologies (Coralville, IA, USA) without introns codon-optimized and cloned into the episomal vector pBGP1 (a kind gift from Charles Lee) 51 using uracil excision-based

cloning as described previously 48 using the primer pair FW-USER-AN4663-episomal and RV-USER-AN4663-GAP-episomal.

For *K. phaffii*, all vectors were linearized before genomic integration using the primers Fw-PmeI_F and Rv-PmeI and the linearized fragments were confirmed by electrophoresis prior to electroporation.

The mixture was centrifuged for 1 minute at 11,000 g to remove cellular debris, and 5 μ L of the supernatant was used as a template for the PCR reaction using OneTaq[®] 2X Master Mix with Standard Buffer (New England Biolabs) primers Fw-AOX1 and Rv-AOX1.”

Primers for K. phaffii	
Fw-PmeI	AAACGCTGTCTTGG AACCTA
Rv-PmeI	AAACTGTCAGTTTTGGGCCAT
Fw_SapI_AN4663	CGGGTTTGCTCTTCGCATACAACGACCGCACTTAATAATATCATC GCTG
Rv_SapI_AN4663	AAACCCGGGGCTCTTCGTTACCAGCCCTCCAGACGCG
FW-NativeSP_LsAA9A	ACGGCACTUTCCTTCGTCGCGTCAGCCGCGGCTCATACCCCTCGTC TGGGGC
RV-NativeSP_LsAA9A	AAGTGCCGUGAGCCCGAGGATGGAGTACTTCATCGTTTGGATCC TTCGAATAATTAGTTGTTTTTGGATC
FW-USER-AN4663-episomal	ATTGAACAACUATTTGAAACGATGGCCCTTTTCGTTCAATTTAT G
RV-USER-AN4663-GAP-episomal	AGTTGTTCAAUTGATTGAAATAGGGACAAATAAATTAATTTAA GTC
Fw-AOX1	GACTGGTTCCAATTGACAAGC
Rv-AOX1	GCAAATGGCATTCTGACATCC

25. Line 548: explain the abbreviation AZCL-HEC.

We have added an explanation for AZCL-HEC and added a reference for the assay from Kracun et al (2015) titled “A new generation of versatile chromogenic substrates for high-throughput analysis of biomass-degrading enzymes” at line 548. It now reads as the following:

“The activity of secreted LsAA9A from *K. phaffii* was determined using a azurine cross-linked hydroxyethylcellulose (AZCL-HEC) chromagenic assay that enables rapid detection of enzymatic activity on polysaccharides⁵³. 1 mg/mL AZCL-HEC substrate (Megazyme, County Wicklow, Bray, Ireland) was mixed with...”

26. The legends of Figures S5-S9 needs serious editing. I use the legend of Fig. S5 as an example. It says: “SDS- PAGE of secreted samples into the supernatant from the co-expression of LsAA9A and the methyltransferase AN4663 when the LsAA9A is expressed using the α -MF signal peptide in biological triplicates.” That is terrible, “lazy” writing. The information provided in the subsequent sentence between parenthesis varies between the strains and is, thus, inconsistent and confusing (e.g., “LsAA9A secreted into the supernatant”). I do not

understand why the arrow indicates an “expected” position in the gel. What is this expectation based on?

We agree with the reviewer that the supplementary text needs to be improved. We have therefore edited the supplementary figure legends accordingly from:

“Supplementary Figure S5 – LsAA9A secretion with the α -MF signal peptide. SDS-PAGE of secreted samples into the supernatant from the co-expression of LsAA9A and the methyltransferase AN4663 when the LsAA9A is expressed using the α -MF signal peptide in biological triplicates. WT (wild-type, parent strain GS115), NHMT (N-terminal histidine methyltransferase expressed from an episomal plasmid), LsAA9A (LsAA9A secreted into the supernatant), LsAA9A + NHMT (LsAA9A co-expressed with the N-terminal histidine methyltransferase). MW (molecular weight in kDa ladder). LsAA9A expected molecular weight without glycosylation is 25.2 kDa. The arrow indicates where the secreted LsAA9A is expected.

Supplementary Figure S6 – LsAA9A secretion with the Amy signal peptide. SDS-PAGE of secreted samples into the supernatant from the co-expression of LsAA9A and the methyltransferase AN4663 when the LsAA9A is expressed using the Amy signal peptide in biological triplicates. WT (wild-type, parent strain GS115), NHMT (N-terminal histidine methyltransferase expressed from an episomal plasmid), LsAA9A (LsAA9A secreted into the supernatant), LsAA9A + NHMT (LsAA9A co-expressed with the N-terminal histidine methyltransferase). MW (molecular weight in kDa ladder). LsAA9A expected molecular weight without glycosylation is 25.2 kDa. The arrow indicates where the secreted LsAA9A is expected.

Supplementary Figure S7 – LsAA9A secretion with its native signal peptide. SDS-PAGE of secreted samples into the supernatant from the co-expression of LsAA9A and the methyltransferase AN4663 when the LsAA9A is expressed using its own native signal peptide in biological triplicates. WT (wild-type, parent strain GS115), NHMT (N-terminal histidine methyltransferase expressed from an episomal plasmid), LsAA9A (LsAA9A secreted into the supernatant), LsAA9A + NHMT (LsAA9A co-expressed with the N-terminal histidine methyltransferase). MW (molecular weight in kDa ladder). LsAA9A expected molecular weight without glycosylation is 25.2 kDa. The arrow indicates where the secreted LsAA9A is expected.

Supplementary Figure S9 – Tf10A secretion with the Amy signal peptide. SDS-PAGE of secreted samples into the supernatant from the co-expression of Tf10A and the methyltransferase AN4663 when the Tf10A is expressed using the Amy signal peptide in biological triplicates. WT (wild-type, parent strain GS115), NHMT (N-terminal histidine methyltransferase expressed from an episomal plasmid), Tf10A (Tf10A secreted into the supernatant), Tf10A + NHMT (Tf10A co-expressed with the N-terminal histidine methyltransferase). MW (molecular weight in kDa ladder). Tf10A expected molecular weight without glycosylation is 21.3 kDa. The arrow indicates where the secreted Tf10A is expected.”

To the following:

Supplementary Figure S5 – SDS-PAGE analysis of α -MF signal peptide-LsAA9A secretion in *K. phaffii*. MW, Marker in kDa. WT, GS115 parent strains. NHMT, strains carrying the episomal plasmid expressing the methyltransferase. LsAA9A, strains expressing LsAA9A with the α -MF signal peptide. LsAA9A+NHMT, strains co-expressing LsAA9A with the α -MF signal peptide and the methyltransferase. The molecular weight of LsAA9A is 25.2kDa. Due to glycosylation from the host *K. phaffii*, the secreted LsAA9A is found at around 37kDa.

Supplementary Figure S6 – SDS-PAGE analysis of Amy signal peptide-LsAA9A secretion in *K. phaffii*. MW, Marker in kDa. WT, GS115 parent strains. NHMT, strains carrying the episomal plasmid expressing the methyltransferase. LsAA9A, strains expressing LsAA9A with the α -MF signal peptide. LsAA9A+NHMT, strains co-expressing LsAA9A with the α -MF signal peptide and the methyltransferase. The molecular weight of LsAA9A is 25.2kDa. Due to glycosylation from the host *K. phaffii*, the secreted LsAA9A is found at around 37kDa.

Supplementary Figure S7 – SDS-PAGE analysis of Native signal peptide LsAA9A secretion in *K. phaffii*. MW, Marker in kDa. WT, GS115 parent strains. NHMT, strains carrying the episomal plasmid expressing the methyltransferase. LsAA9A, strains expressing LsAA9A with the α -MF signal peptide. LsAA9A+NHMT, strains co-expressing LsAA9A with the α -MF signal peptide and the methyltransferase. The molecular weight of LsAA9A is 25.2kDa. Due to glycosylation from the host *K. phaffii*, the secreted LsAA9A is found at around 37kDa.

Supplementary Figure S9 – SDS-PAGE analysis of TfAA10A secretion in *K. phaffii*. MW, Marker in kDa. WT, GS115 parent strains. NHMT, strains carrying the episomal plasmid expressing the methyltransferase. Tf10A, strains expressing TfAA10A with the Amy signal peptide. Tf10A+NHMT, strains co-expressing TfAA10A with the Amy signal peptide and the methyltransferase. The molecular weight of TfAA10A is 21.3kDa. Due to glycosylation from the host *K. phaffii*, the secreted TfAA10A is found at around 25kDa.

The arrow was intended to help visualize where the secreted LsAA9A is found on the SDS-PAGE. The molecular weight of LsAA9A is 25.2kDa according to amino acid composition. However, due to glycosylation, secreted LsAA9A in *K. phaffii* presents a molecular weight of around 37kDa. We have clarified this further in the text.

27. Table S2. Is this an AA10 or an AA13? Some explanatory text may be useful.

The protein is an AA13 family as determined previously and annotated by the CAZy database. The domain predictions in Table S2 are based on pFam which has different classification for the domains and listed simply as “LPMO”. We have modified the text to explain that the glycoside hydrolase domain considered part of the AA13 family of LPMOs by CAZy. The legend text now contains the following addition at the end of the figure legend text:

“The protein is considered part of the AA13 family of LPMOs by CAZy.”

28. Tables S1 & S2. Why are similar Tables not provided for the other two proteins (including the one that was used in the knock out studies aimed at finding the NHMT)?

As mentioned above, there are supplementary for these two proteins because those were the only ones that pFam could accurately predict to contain glycohydrolase domain or classified as AA9 type protein (as in the case for C8V530). We have added more information on the proteins and added 2 extra references to support the statements in the main text.

29. Note S1 and S2: Please indicate if these are the gene sequences as found in the host organism or those synthesized for protein expression in *Pichia*

We thank the reviewer for bringing this to our attention. The gene sequences in Note S1 and S2 are the DNA sequences as found in the host organism, and the ones we used for protein expression in *K. phaffii*. For better clarification, we have changed the legend from:

“Note S1 - DNA sequence LsAA9A

Note S2: DNA sequence TfAA10A”

To:

Note S1: Native DNA sequence of LsAA9A used for heterologous expression in *K. phaffii*.

Note S2: Native DNA sequence of TfAA10A used for heterologous expression in *K. phaffii*.

Some additional grammatical issues:

Lines 55-56: Rephrase

We have rephrased to:

“Histidine methylation^{5,6} on fungal lytic polysaccharide monooxygenases (LPMOs) is particularly unique as this methylation occurs only on the N-terminal histidine that is part of the conserved active site⁷.”

Line 71: fungi-> fungus

We have corrected fungi to fungus.

Line 166: “the process required” makes no sense

We have reworded that sentence to the following:

“Decision flowchart describing the experimental requirements from the PRM assays to nominate a MT candidate to a possible NHMT”

Line 247: are -> is

We thank the reviewer for noticing this grammatical error and have corrected it at this position.

Line 255-257 & 260-262: Rephrase

We have rewritten the entire section as recommended (see above).

Lines 353 – 360: Bad writing. Polish.

We have reworded the second paragraph (starting at line 353) in the discussion from:

“We determined that *nhmT* encodes an NHMT enzyme that shares the seven- β -strand fold with the human histidine methyltransferases METTL9 and METTL18 as well as the human N-terminal methyltransferase METTL13. NHMT also contains an integral membrane region with a potentially novel seven-transmembrane with little structural similarities when compared to known membrane proteins the PDB database. The transmembrane is required for NHMTs specific activity. The 7TM region is unique to filamentous fungi and genes with similar sequence similarity are found primarily as single copies in the genomes of filamentous fungi, suggesting distinct phylogeny of the enzyme.”

To the following:

“The NHMT enzyme shares a seven- β -strand fold in the soluble domain similar to human histidine methyltransferases such as METTL9, METTL18, and the human N-terminal methyltransferase METTL13. Uniquely, NHMT contains an N-terminal seven-transmembrane domain with little structural similarities compared to known membrane proteins in the PDB database. Based on our experimental data, the transmembrane domain is required for the NHMTs specific activity in the context of N-terminal histidine methylation of secreted LPMOs. The sequence of the 7TM region is unique to filamentous fungi and similar sequences are primarily found as single copies in the genomes of filamentous fungi, suggesting distinct phylogeny of the enzyme.”

Reviewer #3 (Remarks to the Author):

Response to Author,

Congratulation on your study. I appreciate the effort in detail work, with CRISPR/Cas9 gene knockout, to gain insights into NHMT biological role and location. The research methods used in the study were appropriate and executed well with details description in the supplement section. The manuscript is well written; clear, and easy to understand.

We thank the reviewer for the positive feedback and greatly appreciate the suggestions and comments to improve the manuscript.

Some minor suggestions:

1. Line 222: add amino acid abbreviations - glutamine 340

We have clarified in the entire text and changed glutamine to glutamic acid 340. In this instance we have added “E340” abbreviation at line 222.

2. Line 658: add percentile of acetonitrile in buffer B

We have added the information on percent of acetonitrile in buffer B, which was 99.9%. We have amended the sentence to the following:

“Buffer A, consisting of 5% acetonitrile with 0.1% formic acid, and buffer B, consisting of 0.1% formic and 99.9% acetonitrile, were used for separation.”

3. Author run two method 21.5 vs 44 minutes for peptide sequencing analysis after fractionation. Any recommendations for the reader which method is better?

We thank the reviewer for noting this difference. Strictly speaking, the 44 minutes method is slightly better and will lead to overall higher number of peptide identifications from a fractionated sample. However, the gains in the number of protein groups is marginal. Practically speaking, 44 minute method is better, however if one is limited by mass spectrometry time, a 21.5 minute method of fractionated peptide mixture will be sufficient for in-depth proteome analysis.

4. Supplement figure 6 and 7. Author claims that molecular weight is around 25.2kDA, however the bands on the gel are around 37kDA for LsAA9A with or without NHMT.

We thank the reviewer for this comment. The molecular weight of the LsAA9A is 25.2kDa without posttranslational modifications. However, the yeast *K. phaffii* glycosylates secreted proteins which consequently increases the molecular weight of the protein. In the case of LsAA9A, the added glycosylation sites from the yeast *K. phaffii* protein production result in a molecular weight of 37kDa size.

We agree that this is a confusing statement and we have therefore amended the supplementary text to better clarify it:

"The molecular weight of LsAA9A is 25.2kDa. Due to glycosylation from the host *K. phaffii*, the secreted LsAA9A is found at around 37kDa."

5. Supplement figure 7 - replicate 1, column 5 (LsAA9A + NHMT), it looks like molecular marker. Please provide better gel.

We thank the reviewer for this comment. While we agree that one of the three replicates (replicate 1, column 5) looks like a molecular marker, unfortunately that is exactly how that specific biological replicate supernatant looked in this instance. Nonetheless, the secreted LsAA9A could be identified from this sample using in-gel mass spectrometry, and these specific samples (in Supplementary Figure 7) were the same ones we used to confirm N-terminal methylation of LsAA9A as presented in Figure 6 (Figure 5 in the revised manuscript) in the main text. To be faithful to the data, we selected to provide the gel for the samples we used to identify the LsAA9A n-terminal histidine methylation.

6. Gene knockout PRM assay supplement figure 1 - combine methylated (609.8351 m/z) and unmethylated (602.8273 m/z) HTVIVYPGYR peptide into one graph.

We thank the reviewer for this suggestion and have made the figure into one figure (see below). We have also changed the figure legend to reflect Figure S1A and S1B.

7. Supplementary Data Set 5, contains PRM quantification used to generate graph in Figure 4A, B. However, figure 4 shows: "Motif and structure and activity prediction from the sequence of NHMT". Please correct.

We thank the reviewer for noting this error. This was due to a typo and we meant Figure 5A, B originally. However, as suggested by one of the other reviewers, we have removed Figure 3 entirely from the manuscript, and moved some aspects to the supplementary data. Therefore, we have one less figure in the main text now and as a consequence of this, the original Figures 4, 5, and 6 in the main text are now Figure 3, 4, and 5 after revision. Hence, Figure 5 in the original version is now Figure 4, so the reference is now correctly matched to the Figure in the main text.

8. Supplement figure 8 contains two out of three calculations, concentrated-LsAA9_MTn is missing, please provide rational why it was not included.

We thank pointing out this emission and mismatch between the supplementary figure and supplementary data table 6. The missing sample info for "concentrated-LsAA9_MTn" was because we originally attempted to express two NHMT (AN4663) DNA sequences in *K. phaffi*: the original native AN46630 DNA sequence from *A. nidulans* with introns (Concentrated-LsAA9_MTn) and a modified sequence in which we removed the introns (Concentrated-LsAA9_MTe). We decided to not include this data in the Supplementary Figure 8 because we could not observe LsAA9A methylation when using the construct with introns (Concentrated-LsAA9_MTn) with the correct signal peptides. We only observed methylation of the LsAA9 protein when expressed from the DNA construct

without introns. We speculate that *K. phaffi* most likely does not splice the introns in the same way as *A. nidulans* and other *Aspergillus* or filamenti fungi strains. As this was beyond the scope of this paper and not discussed in the text we did not visualize the data in Supplementary Figure S8. We have updated Supplementary Table 6 to not include Concentrated-LSAA9_MTn LsAA9A and removed data from samples which are not mentioned or otherwise discussed in the text. We have also included the N-terminal peptide quantification of secreted LsAA9A supplementary table 6. We have also transferred the N-terminal peptide quantification data of secreted TfAA10A from Supplementary table 7 to the new Supplementary table 6. We therefore no longer have seven Supplementary tables.

Reviewer #4 (Remarks to the Author):

This is an interesting work of applying extensive experimental works to identify and characterize a new N-terminal histidine methyltransferase (NHMT) encoded by the gene AN4663 that methylate Lytic polysaccharide monoxygenases (LPMOs). The experiments such as proteomics experiments and gene knockout experiments are well designed and convincingly determine the function and localization of NHMT. The bioinformatics analysis of the protein structure and function such as AlphaFold structure prediction is reasonable. Overall, the methods are solid, and the results are interesting.

We thank the reviewer for the positive summarization and feedback, and we are grateful for the comments.

Comments for revision:

The bioinformatics methods for predicting the structure of NHMT, searching the predicted structure against known structures, constructing multiple sequence alignments for NHMT, and building phylogenetic trees for NHMT proteins across multiple species are not described in detail. A section describing the bioinformatics in detail should be added into the Materials and Methods Section. The data and scripts of the bioinformatics analysis should be made publicly available to reproduce the analysis.

We thank the reviewer for noting the lack of details regarding the bioinformatics analysis. We agree that the Materials and Method section did not contain sufficient details regarding the bioinformatics analysis. We have therefore added additional information in the material and methods to include these details. Materials and Methods now contains additional sections for AlphaFold2 modelling, Structural comparison, and Phylogenetic analysis and visualization.

REVIEWERS' COMMENTS

Reviewer #1 (Remarks to the Author):

The authors have completed a nice study and adequately addressed the issues raised by reviewers and it with one minor exception noted here.

With respect to the comprehensiveness of the test of 22 CRISPR KO strains with 22 different MTs knocked out, the authors did not address the full point. The test employed was an assay for the methylation of a N-terminally histidine-methylated *A. nidulans* peptide sequence. The authors now note that two of the KO strains were not assayed which is good to note. The point was actually that the knockout of the methyltransferase in each of these 22 strains was not shown. Was it assumed that the knockout by HR was successful in all 22 strains or was this empirically verified? The efficiency of the CRISPR system may be high, but presumably there are ways for it to fail. A strain in which the target MT was not actually knocked out would produce the same negative assay result as a strain in which the MT was successfully knocked out but was not the MT providing the methylation of this histidine.

As noted previously, the MT that was a hit in the CRISPR screen and that the authors identify as being responsible for the N-terminal histidine methylation that they are studying is well validated in this manuscript so that is not a concern. It would not be justified to say that MTs could be safely fully ruled out by the CRISPR screen, even those KO strains that did produce an assay result, if it is possible that some of the strains did not have the MT knocked out. A negative result in this assay is exactly the same as the result expected is no KO was achieved in the strain. It's detail in the context of this study's focus, but should be portrayed accurately at the point that this screen is presented. In other words, the screen for MTs with this activity was able to identify a hit and the candidate that they pursued, but the screen alone should not be characterized in the text as a confident rule-out of the other MTs without the empirical confirmation of the knockout of each MT.

Reviewer #2 (Remarks to the Author):

It was difficult to assess the revised version in an efficient manner because this reviewer was not provided with a version in which changes were tracked, whereas the rebuttal letter refers to such a tracked version.

Looking at the rebuttal letter, I believe that the authors have responded very well to my comments and, as far as I can see, the comments from the other reviewers.

There are a few minor issues left that need attention:

1. Line 96-102: The use of the terms "glycoside hydrolase" and the like is confusing. I suggest not to use this term or to explain why this term pops up in some databases. Current phrasing such as ".....and similarly contains a glycoside hydrolase domain classified as AA13 family of LPMOs.....", must be improved/explained. And another example: "and it is classified as a member of the Auxiliary Activity Family 9 (AA9 LPMO) glycohydrolase by CAZY". This is just wrong, very wrong. I find it remarkable that factual errors and badly phrased sentences end up in a revised manuscript when authors even claim that they have carefully revised grammar.

2. In my original review I commented on Fig. 2D,E where the charges are incorrect. The authors responded by explaining the principles of proteomics (well known to this reviewer, who teaches proteomics), claiming that there is no mistake. Fig 2D assigns +2 to the 406 species and Fig 2E assigns +3 to the 602 species. This makes no sense to me. It must be the other way around (as the authors actually write in their rebuttal). What am I overlooking? Or am I right?

REVIEWERS' COMMENTS

Reviewer #1 (Remarks to the Author):

The authors have completed a nice study and adequately addressed the issues raised by reviewers and it with one minor exception noted here.

We would like to thank the reviewer for the input and helping us improve our manuscript considerably.

With respect to the comprehensiveness of the test of 22 CRISPR KO strains with 22 different MTs knocked out, the authors did not address the full point. The test employed was an assay for the methylation of a N-terminally histidine-methylated *A. nidulans* peptide sequence. The authors now note that two of the KO strains were not assayed which is good to note. The point was actually that the knockout of the methyltransferase in each of these 22 strains was not shown. Was it assumed that the knockout by HR was successful in all 22 strains or was this empirically verified? The efficiency of the CRISPR system may be high, but presumably there are ways for it to fail. A strain in which the target MT was not actually knocked out would produce the same negative assay result as a strain in which the MT was successfully knocked out but was not the MT providing the methylation of this histidine.

As noted previously, the MT that was a hit in the CRISPR screen and that the authors identify as being responsible for the N-terminal histidine methylation that they are studying is well validated in this manuscript so that is not a concern. It would not be justified to say that MTs could be safely fully ruled out by the CRISPR screen, even those KO strains that did produce an assay result, if it is possible that some of the strains did not have the MT knocked out. A negative result in this assay is exactly the same as the result expected if no KO was achieved in the strain. It's detail in the context of this study's focus, but should be portrayed accurately at the point that this screen is presented. In other words, the screen for MTs with this activity was able to identify a hit and the candidate that they pursued, but the screen alone should not be characterized in the text as a confident rule-out of the other MTs without the empirical confirmation of the knockout of each MT.

We thank the reviewer for the clarification, and we apologize that we were not able to address the core question in our initial response. As the reviewer notes, it is not well explained how we determined if the knockouts were successful. We indeed validate our knockouts with diagnostic tissue-PCR and the point mutations and RFP-tagging were validated using Sanger-sequencing. Following the reviewer's advice, we have added this information (underlined below) to line 148 in the text to highlight that the KOs were validated with the following:

“CRISPR/Cas9-mediated gene knockouts were confirmed by diagnostic PCR and to analyse the effect of the individual knockouts on N-terminal histidine methylation, we developed a targeted proteomics assay based on parallel reaction monitoring (PRM) to specifically monitor and quantify the native N-terminally histidine methylated *A. nidulans* peptide sequences identified in the proteome analysis (Figure 2A).”

We have also modified the methods section to describe that sanger-sequence was utilized for the validation of point mutations and RFP-tagging. Thus lines 419 – 430 now read the following:

“Protoplastation for *A. nidulans* was performed as previously 50. The transformations were made in gently thawed protoplasts²³. Each transformation protoplasts were mixed with 1.5 µg of CRISPR vector and used in combination with either 2 µg of a linear double strand DNA for homologous recombination or 20 µL of 100 µM stock solutions of oligo nucleotides in a total volume of 150 µL of PCT buffer (50 % w/v PEG8000, 50 mM CaCl₂, 20 mM Tris, 0.6 M KCl, pH 7.5). The mix was incubated for 10 minutes at room temperature, followed by addition of 250 µl of transformation buffer (1.2 M sorbitol, 50 mM CaCl₂·2 H₂O, 20 mM Tris, 0.6 M KCl, pH 7.2), and plated on transformation media (1M sucrose, 2% agar, 1 x nitrate salt solution, 0.001% thiamine, 1 x trace metal solution plates). All transformation plates were incubated at 37°C. Resulting transformants were examined by diagnostic tissue-PCR as described previously²². Sanger-sequencing was used for validation of strains with AN4663 complementation, point-mutation, and RFP-tagging.”

Reviewer #2 (Remarks to the Author):

It was difficult to assess the revised version in an efficient manner because this reviewer was not provided with a version in which changes were tracked, whereas the rebuttal letter refers to such a tracked version.

Looking at the rebuttal letter, I believe that the authors have responded very well to my comments and, as far as I can see, the comments from the other reviewers.

We thank the reviewer for the constructive input that allowed us to improve the manuscript.

There are a few minor issues left that need attention:

1. Line 96-102: The use of the terms “glycoside hydrolase” and the like is confusing. I suggest not to use this term or to explain why this term pops up in some databases. Current phrasing such as “.....and similarly contains a glycoside hydrolase domain classified as AA13 family of LPMOs.....”, must be improved/explained. And another example: “and it is classified as a member of the Auxiliary Activity Family 9 (AA9 LPMO) glycohydrolase by CAZy”. This is just wrong, very wrong. I find it remarkable that factual errors and badly phrased sentences end up in a revised manuscript when authors even claim that they have carefully revised grammar.

The reviewer is indeed correct that the term “glycoside hydrolase” and the like, are confusing. We agree that the classifications in all the databases are perhaps not up to date. Much is due to legacy classification of protein families such as those of Auxiliary Activity Family 9 which were originally placed in the glycoside hydrolases family 61 (GH61) due to low endoglucanase activity but are now in a separate family known as lytic polysaccharide monoxygenases. However, not all databases such as pFam have updated the classifications in their databases and are always evolving and thus we did not elaborate on this for clarification. To address the comment, we have removed the usage of term “glycoside hydrolase” from the text and now updated the relevant section to the following:

“C8V530 (gene ID AN10419), is classified as a member of the Auxiliary Activity Family 9 (AA9) of lytic polysaccharide monooxygenases(LPMOs, Table S1) 15–17. Q5B1W7 (AN5463) plays a crucial role in starch degradation 18, and similarly classified as AA13 family of copper-dependent LPMOs shown to be active on starch¹⁹ (Table S2). Q5AU55 (AN4702) is structurally similar to an AA11-type LPMO from *A. oryzae*. Lastly, Q5B428 (AN8175) is highly homologous to an LPMO-like protein from *Laetisaria arvalis* (termed LaX325), part of the recently defined protein family “X325” that are found to be widespread in the fungal kingdom but have deviated evolutionary and do not perform oxidative cleavage of polysaccharides²⁰. All four of these LPMO-related proteins contain a secretion signal peptide prior to the N-terminal histidine of the processed proteins.”

2. In my original review I commented on Fig. 2D,E where the charges are incorrect. The authors responded by explaining the principles of proteomics (well known to this reviewer, who teaches proteomics), claiming that there is no mistake. Fig 2D assigns +2 to the 406 species and Fig 2E assigns +3 to the 602 species. This makes no sense to me. It must be the other way around (as the authors actually write in their rebuttal). What am I overlooking? Or am I right?

The reviewer is indeed correct. We apologize for the confusion as we mistakenly thought the reviewers' concerns were regarding another aspect of Figure 2. The charge states of Figure 2D and E are flipped. This was due to an error during editing of the figure. It has been fixed in the final version.